



**Importance of the information content in the study area when regionalising rainfall-runoff**
**model parameters: the role of nested catchments and gauging station density**
Mattia Neri[1], Juraj Parajka[2], Elena Toth[1]
[1] DICAM, University of Bologna, Bologna, Italy
[2] Institute for Hydraulic and Water Resources Engineering, Vienna University of Technology, Austria
*Correspondence to*: Mattia Neri (mattia.neri5@unibo.it)
**Abstract.**
The set up of a rainfall-runoff model in a river section where no streamflow measurements are available for its
calibration is one of the key research activity for the Prediction in Ungauged Basins (PUB): in order to do so it is
possible to regionalise the model parameters based on the information available in gauged sections in the study region.
The information content in the data set of gauged river stations plays an essential role in the assessment of the best
regionalisation method: this study analyses how the performances of different model regionalisation approaches are
influenced by the "information richness" of the available regional data set, and in particular by its gauging density and
by the presence of nested catchments, that are expected to be hydrologically very similar.
The research is carried out over a densely gauged dataset covering the Austrian country, applying two different rainfall-
runoff models: a semi-distributed version of the HBV model (TUW model), and the Cemaneige-GR6J model. The
regionalisation approaches include both methods which transfer the entire set of model parameters from donor
catchments, thus maintaining correlation among parameters ("output averaging" techniques), and methods which derive
each target parameter independently, as a function of the calibrated donors' ones ("parameter averaging" techniques).
The regionalisation techniques are first implemented using all the basins in the dataset as potential donors, showing that
the output-averaging methods outperform the parameter-averaging kriging method, highlighting the importance of
maintaining the correlation between the parameter values.
The regionalisation is then repeated decreasing the information content of the data set, by excluding the nested basins,
identified taking into account either the position of the closing section along the river or the percentage of shared
drainage area. The parameter-averaging kriging is the method that is less impacted by the exclusion of the nested
donors, whereas the methods transferring the entire parameter set from only one donor suffer the highest deterioration,
since the single most similar or closest donor is often a nested one. On the other hand, the output-averaging methods
degrade more gracefully, showing that exploiting the information resulting from more than one donor increases the
robustness of the approach also in regions that do not have so many nested catchments as the Austrian one.
Finally, the deterioration resulting from decreasing the station density on the regionalisation was analysed, showing that
the output averaging methods using as similarity measure a set of catchment descriptors, rather than the geographical
distance, are more capable to adapt to less dense datasets.
The study confirms how the predictive accuracy of parameter regionalisation techniques strongly depends on the
information content of the dataset of available donor catchments and indicates that the output-averaging approaches,
using more than one donor basin but preserving the correlation structure of the parameter set, seem to be preferable for
regionalisation purposes in both data-poor and data-rich regions.





## 1 Introduction

In the hydrological practice, it is often needed to gain information on ungauged river sections and one of the most informative way to do so is implementing a rainfall-runoff model, when, as it is often the case, the meteorological input variables are retrievable in reference to its drainage area. Since in such cases the model parameters may not be obtained through a calibration procedure, it is necessary to regionalise them, exploiting the information of the hydrometric measurements collected in hydrologically similar catchments in the study area.

Regionalisation approaches for model parameterisation can be classified into two wide categories: "regression-based" methods and "distance-based" methods (He et al., 2011). The former techniques try to define relationships between each model parameter and geomorpho-climatic catchment attributes (see e.g., Seibert 1999). The latter, instead, identify a set of donor watersheds (with similar attributes) and transfer their calibrated parameters to the ungauged ("target") catchment. This last type of approaches includes both methods which transfer the entire set of model parameters from donor catchments, thus maintaining correlation among parameters (also named "output averaging" techniques, which run the model multiple times and average the simulations), and methods which derive each target parameter independently, as a function (generally a weighted average) of the calibrated donors' ones ("parameter averaging" techniques). To the latter class ("distance-based" group of the "parameter averaging" type) belong also the kriging methods, where the parameters are regionalised based on their spatial correlation and independently from each other (Merz and Blöschl, 2004; Parajka et al., 2005).

In the last two decades, hydrologic scientists from all around the word have focused on the determination of the more accurate regionalisation techniques for different case studies and rainfall-runoff models (see e.g., the reviews of Merz et al. 2006, He et al. 2011, Peel & Blöschl 2011, Parajka et al. 2013, Hrachowitz et al. 2013, Razavi and Coulibaly 2013).

A very important aspect for choosing the most adequate regionalisation technique, and that is worthy of further analyses, is the information content of the study region. In particular, in very densely gauged areas, spatial proximity is expected to be a good similarity measure, as demonstrated by the studies by Merz and Blöschl (2004) and Parajka et al. (2005), who tested different regionalisation approaches on a dense dataset of more than 300 watersheds across Austria, and by Oudin et al. (2008), on a set of 913 catchments in France, finding that the techniques based on spatial proximity alone provided excellent performances. But different outcomes may result for less densely and less interconnected (that is with less availability of stations along the same river), as shown for instance, by Samuel et al. (2011): they regionalised the parameters of HBV model for a strongly less densely gauged dataset (135 watersheds on the wide area of Ontario, in Canada) and found that the best the best approach for such study area was an inverse-distance parameter averaging for a pre-selected set of physically similar catchments.

The availability in the data set of gauged river stations representative of hydrological conditions similar to the ungauged ones plays an essential role in the assessment of the best regionalisation method. This availability can be, in some way, estimated with the station density (i.e. number of station per $km^2$) and with the topological relationship between catchments. In particular, the presence of several nested catchments (i.e. gauged river sections on the same river) in the study region can strongly influence the performance of certain techniques: if for an ungauged basin, model parameter sets are available for down/upstream gauged river sections, donor and target watersheds share indeed part of their drainage area, and thus they may be also hydrologically very similar. This may actually lead to very good regionalisation performances for a given approach, but such accuracy may not represent what would be obtained in





different conditions. Therefore, regionalisation performances obtained for datasets with high degree of "nestedness"
may be not transferrable to study regions poor of nested basins.

So far, very few studies have been presented in the literature regarding the impact of the presence of nested catchments
on the performances of parameter regionalisation techniques. Merz and Blöschl (2004), Parajka et al. (2005) and Oudin
et al. (2008) tested the effect of the removal of nested catchments from the available donor catchments, but only for one
or two regionalisation techniques, without analysing in detail the differences between different types of approaches.
Additionally, the contribute of the immediate downstream and/or upstream gauged stations has never been compared to
that of the remaining nested catchments, that may share significant portions of drainage area with the ungauged one.
Also the influence of the density of the gauging stations on the parameterisation of rainfall-runoff models has been little
explored, with two notable exceptions: Oudin et al. (2008) applied the spatial proximity and physical similarity output-
averaging techniques for decreasing values of station density in France and Lebecherel et al. (2016) tested the
robustness of the spatial proximity output-averaging approach to an increasing sparse hydrometric network on the same
study region. In Austria, the effect of station density has been investigated by Parajka et al. (2015), but in reference to
the interpolation of streamflow time-series and not to the parameterisation of rainfall-runoff models.

The purpose of the present paper is to compare the impact of the presence of nested donors on the performances of
different parameter regionalisation techniques for a dataset of 209 catchments across Austria. The effect of nested
donors is here tested for a set of consolidated techniques, applied to two different continuous simulating daily rainfall-
runoff models, for generalisation purposes: the first is the TUW model (semi-distributed version of HBV, used by
Parajka et al. 2005), and the second model, never used so far for regionalisation in the Austrian region, is the GR6J
model (Pushpalatha et al. 2011) implemented with the Cemaneige snow routine (Valery et al. 2014).
For the exclusion of nested basins, we propose two different criteria, taking into account either the position of the
closing section along the river or the percentage of shared drainage area. The results are also compared to the effect of
the reduction of station density, following a procedure similar to what was done, for different purposes, by Parajka et al.

(2015).

We believe that the present analysis may provide further insights for assessing the performances and selecting the
parameter regionalisation approaches most suitable to a specific study region, keeping into account the impact of the
topological information "richness" of the available regional dataset.
The paper is organized as follows: Section 2 introduces the case study and data. Section 3 first describes the rainfall-
runoff models and the tested regionalisation schemes, then the methodology for assessing the impact of nested
catchments and of station density is presented, while the results are presented in Section 4. Finally, Section 5 reports the
discussion and the conclusions.
**2 Study region and data**
The case study is composed by 209 catchments (see Figure 1) covering a large portion of Austria. Their size varies
considerably, from 13 to over 6000 km$^2$. The topography of the country varies significantly from the flat and hilly area
in the north-east to the Alps in the centre and in the south-west, particularly steep in the extreme west. The annual
precipitation ranges from about 600 mm in the east, where the evaporation plays an important role in the water balance,
to the more than 2000 mm in the west, mainly due to orographic lifting of north-westerly airflows at the rim of the Alps





(Viglione et al., 2013). Land use is mainly agricultural in the lowlands and forest in the medium elevation ranges.
Alpine vegetation and rocks prevail in the highest catchment (Parajka et al., 2005). The aridity index assumes values
from 0.2 to 1, meaning that the watersheds are mainly wet or weakly arid (annual evapotranspiration is never higher
than precipitation).
Data have been provided by the Institute of Hydraulic Engineering and Water Resources Management (Vienna
University of Technology), which previously screened the runoff data for errors and removed all stations with
significant anthropogenic effects. Hydro-meteorological data include daily streamflow and daily inputs to the rainfall-
runoff models for the 33 years period 1976-2008: daily average precipitation, temperature and potential
evapotranspiration defined for 200 meters elevation zones for all the study catchments. The potential evapotranspiration
is estimated by a modified Blaney-Criddle method (Parajka et al., 2005) using interpolated daily air temperature and
grid maps of potential sunshine duration (Mészároš et al., 2002).
In order to implement some of the parameter regionalisation approaches, we make use of several geo-morphoclimatic
catchment attributes, reported and briefly described in Table 1. Topographic attributes such as mean catchment
elevation and mean slope are derived from 1 x 1 km digital elevation model while climatic features such as mean annual
precipitation, and aridity index are derived from climatic input time series. Figure 2 shows the spatial pattern of mean
annual precipitation, snow depth and aridity index across the study area. Mean annual solar irradiance is computed
trough GRASS GIS software (http://grass.osgeo.org). Stream network density, FARL (flood attenuation by reservoir
and lakes), boundaries of porous aquifers, areal portions of regional soil types and main geological formation were the
same used and described in detail in Parajka et al. (2005). Finally, Land use coverage is derived from CORINE Land
Cover maps updated to year 2012 (https://land.copernicus.eu/pan-european/corine-land-cover/clc-2012). For land cover
classes, as well as for geology and soil type classes, the catchments are associated to more than one single attribute:
each basin is described by the portions of the total catchment area corresponding to each class (and for this reason,
Table 1 does not report the min/median/max values of such descriptors.

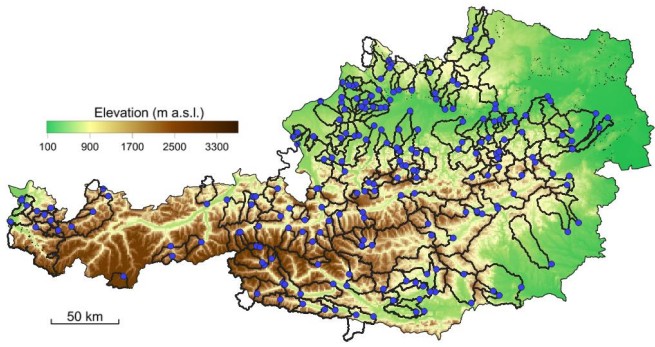

Figure 1. Study area, blue points refer to stream gauges and black lines to catchment boundaries.

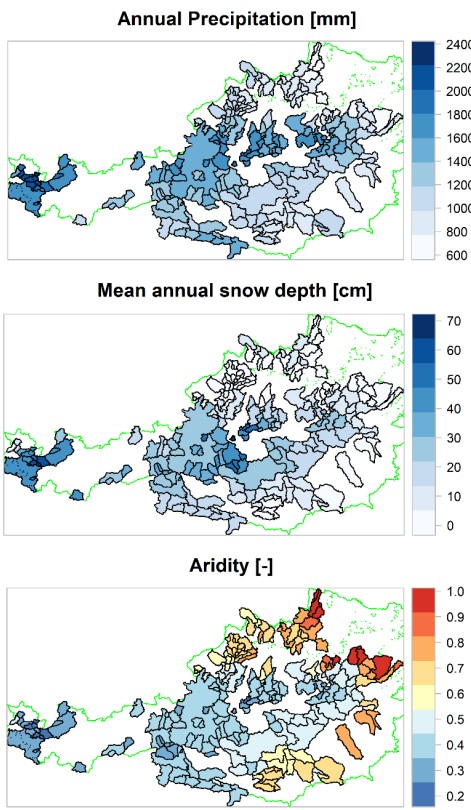

Figure 2. Spatial pattern of some climatic catchment attributes across the study area.
Table 1. Available catchment attributes.

| Code | Unit | Min | Median | Max | Description |
|------|------|-----|--------|-----|-------------|
| Elev | m a.s.l. | 287 | 915 | 2964 | Mean elevation |
| Area | km² | 14 | 168 | 6214 | Drainage area |
| Slope | m/m | 0.9 | 12.4 | 28.5 | Mean slope |
| meanP | mm | 675 | 1230 | 2310 | Mean annual total precipitation |
| maxP | mm | 35 | 49 | 84 | Mean annual maximum daily precipitation |
| meanPET | mm | 281 | 608 | 715 | Mean annual total evapotranspiration |
| SnowF | - | 0.06 | 0.17 | 0.60 | Fraction of precipitation fallen as snow (i.e. precipitation fallen in days below 0°) |
| SnowD | mm | 1 | 14 | 68 | Mean annual snow depth |
| Aridity | - | 0.21 | 0.46 | 0.96 | Aridity index (meanPET/meanP) |
| Irrad | kWh/(m²*day) | 1750 | 1899 | 2274 | Mean annual solar irradiance |
| RiverD | m/km⁻² | 0 | 830 | 1256 | Stream network density |
| FARL | - | 0.56 | 1 | 1 | Flood attenuation index by reservoir and lakes |
| Corine | % | - | - | - | Portions of land use coverage |
| Geology | % | - | - | - | Portions of geological formations |
| Soils | % | - | - | - | Portions of regional soil types |
| Forest | - | 0 | 0.47 | 0.93 | Fraction of catchment covered in forest |
| AcqPort | - | 0 | 0.01 | 0.83 | Fraction of catchment with porous aquifers |



## 3 Materials and methods

### 3.1 Rainfall-runoff models structure and calibration

Two models for simulating daily streamflow were applied in this study. This choice is made in order to analyse the effect of nested catchments and station density on the performance of parameter regionalisation methods for different model structures.

### 3.1.1 TUW model

The first is the TUW model, a semi-distributed version of the HBV model (Bergström 1976, Lindström et al., 1997) developed by Parajka and Viglione (2019). It consists in a snow routine, a soil moisture routine and a flow response and routing routine. The model processes the elevation zones as autonomous entities that contribute separately to the total outlet flow. The inputs are daily air temperature, precipitation and potential evapotranspiration over the different elevation zones, on which the model is run in the version schematized in Figure 3. Finally, the different outputs from the elevation zones are averaged taking into account the sub-catchment areas.

The snow routine is based on a simple degree-day concept and it is ruled by five parameters: two threshold temperature parameters distinguishing rain and snow, $Tr$ and $Ts$, a melting temperature $Tm$, a snow correction factor $SCF$ and the degree-day factor $DDF$. The soil moisture routine represents soil moisture state changes and runoff generation and involves three parameters: the maximum soil moisture storage $FC$, a parameter representing the soil moisture state above which evapotranspiration is at its potential rate, $LP$, and a parameter $\beta$ ruling the non-linear function of runoff generation. Finally, an upper and a lower soil reservoirs and a triangular transfer function compose the runoff response and routing routine, involving seven additional parameters. The sum of excess rainfall and snowmelt enters the upper zone reservoir and leaves this reservoir through three paths: i) outflow from the reservoir based on a fast storage coefficient $k_1$; ii) percolation to the lower zone with a constant percolation rate $C_{perc}$, iii) if a threshold of the upper storage state $L_{UZ}$ is exceeded, through an additional outlet based on a very fast storage coefficient $k_0$. Water leaves the lower zone based on a slow storage coefficient $k_2$. The outflows from both reservoirs are then routed by a triangular transfer function representing runoff routing in the streams, where the base of transfer function, $B_Q$, is estimated with the scaling of the outflow by the $C_{ROUTE}$ and $B_{MAX}$ parameters. More details about the model structure and application in R can be found in Parajka et al. (2007) and Ceola et al (2015), respectively.

The model is run for all the study catchments with the semi-distributed model structure obtained dividing them into 200-meters elevation zones: model daily inputs (precipitation, temperature and potential evapotranspiration) and model states are defined over such zones, while model parameters are assumed to be the same for the entire catchment.

Following the work by Parajka et al. (2005) on the same study area, 4 out of the 15 total parameters are pre-set and 11 are calibrated: threshold temperatures $Tr$ and $Ts$ are fixed respectively to 2 and 0 °C, $Tm$ to 0 °C and the maximum base of the transfer function at low flows $B_{MAX}$ to 10 days. Table 2 briefly reports and describes the calibrated parameters, defining also their lower and upper bounds.



Table 2. TUW model parameters and their ranges.

| Parameter | Units | Range | Description |
|---|---|---|---|
| SCF | - | 0.9 - 1.5 | Snow correction factor |
| DDF | mm/(°C*day) | 0 - 5 | Degree day factor |
| LP | - | 0 - 1 | Parameter related to the limit of evaporation |
| FC | mm | 0 - 600 | Field capacity, i.e., max soil moisture storage |
| $\beta$ | - | 0 - 20 | Non linear parameter for runoff production |
| $k_0$ | days | 0 - 2 | Storage coefficient for very fast response |
| $k_1$ | days | 2 - 30 | Storage coefficient for fast response |
| $k_2$ | days | 30 - 250 | Storage coefficient for slow response |
| $L_{UZ}$ | mm | 0 - 100 | Threshold storage state, very fast response starts if exceeded |
| $C_{perc}$ | mm/day | 0 - 8 | Constant percolation rate |
| $C_{ROUTE}$ | days$^2$/mm | 0 - 50 | Scaling parameter |

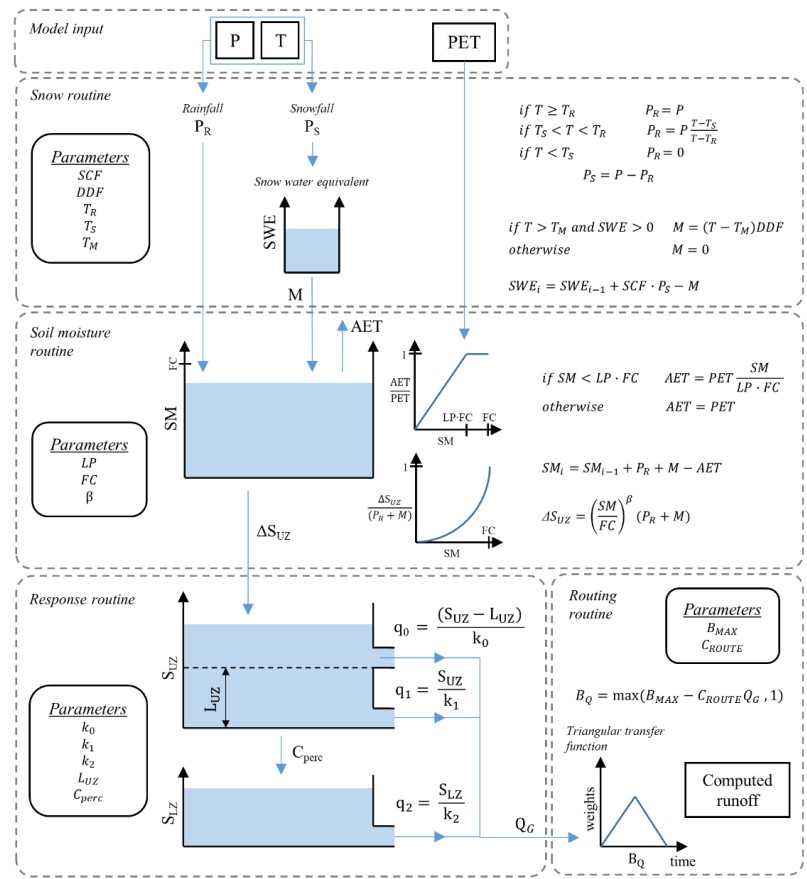

Figure 3. TUW model scheme – Lumped version.





### 3.1.2 CemaNeige-GR6J model

The second model is the French CemaNeige-GR6J. It is the combination of the CemaNeige snow accounting routine (Valéry et al., 2014) with the GR6J model (Pushpalatha et al., 2011), a daily lumped continuous rainfall-runoff model, developed at IRSTEA (Anthony, France), by the Équipe Hydrologie des Bassins versants.

The inputs of the model are spatially-averaged catchment daily air temperature, precipitation and potential evapotranspiration. Catchment hypsometric curve is also required.

The CemaNeige snow accounting routine is based on a degree-day concept, where the thermal inertia of the snowpack is also taken into account. It involves two parameters, a snowmelt factor, $\theta_{G1}$, and a cold-content factor, $\theta_{G2}$. Although the module requires daily lumped inputs, for better simulating snow accumulation and melting it allows to divide the catchment into more elevation zones of equal area, through the use of the hypsometric curve. Inputs for each elevation zone are extracted through interpolation of the mean catchment values using precipitation and temperature gradients (Valéry et al, 2010), and not from "clipping" of the actual spatial fields like for the TUW elevation zones. The module functions are applied with a lumped set of calibrated parameters; but internal states are allowed to vary over each elevation layer according to the different extrapolated inputs. On each elevation layer, two outputs are computed: rain and snowmelt, which are summed in order to find the total water quantity feeding the hydrological model. At every time step, the total liquid output of CemaNeige at catchment scale is the average of every elevation zone outputs. Here we decide to maintain, as default, the number of elevation layers equal to five. For a detailed description of CemaNeige routines, the readers may refer to Valéry et al. (2014).

The total liquid output of CemaNeige module and potential evapotranspiration are the inputs of the GR6J rainfall-runoff model. In the model, the water balance is controlled by a soil moisture accounting reservoir and a conceptual "groundwater" exchange function, while the routing part of the structure consists in two flow components routed by two unit hydrographs, a non-linear store and an exponential-store, with a total of six parameters. The structure of the model is represented in Figure 4 and a detailed description of the model routines is given in Pushpalatha et al. (2011).

The CemaNeige-GR6J model is fed with mean catchment daily precipitation, air temperature and potential evapotranspiration. All the 8 parameters of the combined model (2 for CemaNeige, 6 for GR6J) are calibrated. Lower and upper bounds of the parameters space are kept as default: all the parameters are allowed to vary between the normalized interval [-9.99 9.99] and then specific parameter transformations are applied before the model is run. Table 3 reports brief parameters description and transformed boundaries. For the sake of simplicity, we will refer to this model just with the acronym GR6J, even if it will always include the CemaNeige snow module.

Table 3. Cemaneige-GR6J model parameters and their transformed real ranges.

| Parameter | Units | Range | Description |
|---|---|---|---|
| $\theta_{G1}$ | mm/(°C*day) | 0 - 109 | Snowmelt (degree-day) factor |
| $\theta_{G2}$ | - | 0 - 1 | Cold content factor |
| X1 | mm | 0 - 21807 | Non-linear production storage capacity |
| X2 | mm | -1903 - 1903 | Groundwater exchange coefficient |
| X3 | mm | 0 - 21807 | Non-linear routing store capacity |
| X4 | days | 0 - 22 | Time parameter for unit hydrographs routing |
| X5 | - | 0 - 1 | Threshold parameter for water exchange with groundwater |
| X6 | mm | 0 - 21807 | Exponential routing store capacity |



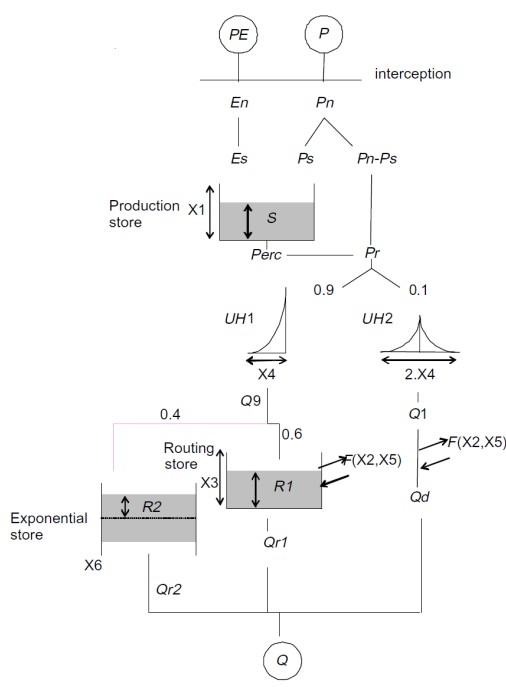

Figure 4. GR6J model scheme.

### 3.1.3 Model calibration

The sets of parameters for both rainfall-runoff models are estimated for all the study catchments with an automatic
model calibration procedure, using the Dynamically Dimensioned Search (DDS algorithm, Tolson et al. 2007).
The objective function to be maximized is the Kling-Gupta Efficiency (Gupta et al., 2009) between observed and
simulated streamflow, defined as:

$$KGE = 1 - \sqrt{(r-1)^2 + (\alpha-1)^2 + (\beta-1)^2} \qquad Eq. 1$$

where $r$ is the Pearson product moment correlation coefficient, $\alpha$ is ratio between the standard deviations of the
simulated and observed values and β is ratio between the means of the simulated and observed values.
The 33 years of observation (1976-2008) are split into two sub-periods: the first one, from 1 November 1976 to 31
October 1992, is used for model calibration, and the second one, from 1 November 1991 to 31 October 2008, for model
validation. Warm-up periods of one year are used in all cases. Calibration and validation performances for both models
are reported in Section 4.1.

### 3.2 Regionalisation approaches

In order to assess the impact of the presence of nested catchments and station density on the performance of the
parameter regionalisation methods, a set of consolidated approaches for the study area are implemented. Three types of
techniques are tested, all belonging to the distance-based group, since recent studies have demonstrated how should be





preferred to regression-based techniques (see e.g. Kokkonen et al. 2003, Merz and Blöschl 2004, Oudin et al. 2008,
Reichl et al. 2009, Bao et al. 2012, Steinschneider et al. 2015, Yang et al. 2018, Cislaghi et al. 2019).

### 3.2.1 Ordinary kriging (KR)

The first is a parameter-averaging technique, based on an ordinary kriging approach (termed in the following KR),
where each model parameter is regionalised independently from each other, based on their spatial correlation.
Catchment position is defined by the coordinates of the catchment centroid and the ordinary kriging is based on an
exponential variogram with a nugget of 10% of the observed variance, a sill equal to the variance, and a range of 60 km
both for TUW and Cemaneige-GR6J model parameters.

### 3.2.2 Nearest Neighbour (1 donor, NN-1)

The second approach is a nearest neighbour method (NN-1), where the complete set of model parameters is transposed
from the geographically nearest donor catchment.

### 3.2.3 Most Similar (1 donor, MS-1)

In the third technique, termed "most similar" approach (MS-1), a single donor catchment is again identified, for
transposing the entire parameter set but, instead of choosing the catchment that is geographically the closest, the
"hydrologically most similar" donor is identified, based on a set of geo-morphological and climatic descriptors. Five
descriptors are used for assessing such similarity: mean catchment elevation, long-term mean annual precipitation,
stream network density, land cover classes, geology classes. Such set of descriptors was selected by preliminary tests:
since it is not the focus of the work, the analysis for the assessment of the best catchment descriptors is reported in
Appendix A. The donor catchment is identified as the catchment with the smallest dissimilarity index $\phi$ (e.g. Burn and
Boorman, 1993):

$$\phi = \sum_{j=1}^{5} \frac{d_j(D,U)}{\max(d_j)} \qquad\qquad Eq.\ 2$$

which represents the sum of the differences $d_j$ of the 5 descriptors of the donor catchment $D$ and of the ungauged
catchment $U$ of interest, normalised by their maximum. For the attributes described by a single value (mean catchment
elevation, long-term mean annual precipitation and stream network density), $d_j$ is expressed by the absolute difference
between the descriptors $X_j^D$ and $X_j^U$ of the donor and target catchments respectively (Eq. 3). For land cover and geology,
whose attributes $X_j$ are the vectors containing the portions of the total catchment area $X_{j,c}$ corresponding to each class $c$,
the difference $d_j$ is calculated as the Euclidean distance between such vectors (Eq. 4).

$$d_j(D,U) = \left| X_j^D - X_j^U \right| \qquad\qquad Eq.\ 3$$

$$d_j(D,U) = \sqrt{\sum_c (X_{j,c}^D - X_{j,c}^U)^2} \qquad\qquad Eq.\ 4$$





### 3.2.4 Output averaging version of NN and MS techniques (NN-OA and MS-OA)

Nearest Neighbour and Most Similar approaches allow to maintain correlation among model parameters, and overcomes the well-known limitation of the regression approach due to interaction between them. In the "regression-based" methods in fact, as well as in the parameter-averaging approaches (e.g, KR technique), parameters are regionalised independently from each other, possibly affecting simulation performances. On the other hand, one single donor catchment (as in NN-1 and MS-1 approaches) is often not fully representative of the hydrological behavior of the target watershed. Recent studies have been demonstrating how averaging the outputs of the simulations (rather than model parameters) obtained with different donor parameter sets may be preferred (see e.g., Oudin et al. 2008, Viviroli et al. 2009). For this reason, NN and MS techniques are also tested identifying more than one donor (here termed NN-OA and MS-OA respectively), with an 'output-averaging' approach (introduced by McIntyre et al., 2005): $n$ donor basins (the geographically closest ones for the nearest neighbour method, or those with the smallest similarity indexes for the "most similar" method) are identified. The regionalised streamflow for the ungauged catchment is calculated from all the simulations $Q(d, P_i)$, obtained running the model (fed by the meteorological input of the target catchment) with each one of the $n$ parameter sets ($P_i$, with $i$ in $[1 ; n]$) corresponding to each of the donor catchments. Streamflow for day $d$, $Q(d)$, is computed as the weighted average of the simulated outputs:

$$Q(d) = \sum_{i=1}^{n} w_i \, Q(d, P_i) \qquad \qquad Eq. 5$$

where $w_i$ is the weight associated to each donor catchment $i$, computed as function of a measure of dissimilarity between the donor and the target catchments. In the NN-OA case, the dissimilarity is defined by the spatial distance $D_i$ between the centroids of donor $i$ and target catchments (Eq. 6), while in the MS-OA method it corresponds to the dissimilarity index $\phi_i$ (Eq. 7).

$$w_i = \frac{\frac{1}{D_i}}{\sum_{i=1}^{n} \frac{1}{D_i}} \qquad \qquad Eq. 6$$

$$w_i = \frac{\frac{1}{\phi_i}}{\sum_{i=1}^{n} \frac{1}{\phi_i}} \qquad \qquad Eq. 7$$

### 3.2.5 Choice of the number of donor catchments for NN-OA and MS-OA

The choice of the number of donor catchments for output averaging represents a central issue in the methodology. Previous studies showed that the optimal number of donors is strongly related to the rainfall-runoff model and, of course, to the case study. McIntyre et al. (2005) were amongst the first to apply an ensemble ("output averaging") approach and to explore the use of different numbers of donors on the performance of the Probability Distribution Model (PDM, Moore, 1985) for a set of more than 100 UK catchments. They tested the impact of an increasing number of donors, either selecting the first $n$ catchments with the smallest dissimilarity measure, or including all the donors with a value of dissimilarity below a defined threshold (in the latter case, the number of donors may thus vary depending on the target-donors attributes). They found that a fixed number of ten donors resulted in the best regionalisation performances. Oudin et al. (2008) applied an output-averaging regionalisation for the TOPMO and GR4J models to a



large French dataset of almost 1000 basins, but with no weights in flow averaging, since they used an arithmetic
average (thus not taking into account magnitude of donor dissimilarities). They found that the two models performed
optimally with a different number of donor catchments (seven and four respectively) and the efficiency of the
regionalised model decreased almost linearly when increasing the number of donors above such values. In fact, the
higher is the number of donor basins included in the regionalisation process, the more dissimilar will be the donors with
respect to the target watershed, possibly leading to a deterioration of the results. The use of weights in flow averaging
may indeed help to smooth this effect, giving less and less importance to the donors as their similarity decreases.
In the present work, the effect on regionalisation performances due to the number of donor basins is explored in detail,
applying NN-OA and MS-OA for increasing number $n$ of donor catchments, as discussed in Section 4.2.

### 3.3 Impact of nested catchments: which catchments should be considered (to be) nested?

As already introduced, the main purpose of the present analysis is to quantify the impact of the presence of several
nested catchments on the regionalisation techniques. In particular, since nested catchments may have a strong
hydrological similarity with the ungauged one, they are expected to play an essential role in the determination of
method performances.
Once the performances have been evaluated using all the study catchments as potential donors, the regionalisation
procedures are repeated for each target basin (assumed to be ungauged) by excluding, from the donors set, the
watersheds which are considered to be nested in relation to the target section.
In general, two or more catchments are nested between each other if their closure sections are located on the same river,
i.e. they share part of their drainage area. Since it may happen that several gauged stations are located on the same river,
we propose to follow two different criteria in order to identify the nested basins:
- *Criterion 1*: the gauged sections that are immediately downstream and upstream of the target section
- *Criterion 2*: all the catchments sharing a given percentage of drainage area with the ungauged one.

### 3.4 Impact of station density

Another way to evaluate the performances of regionalisation methods taking into account the "richness" in hydrometric
information of the study area is to analyse the spatial density of the potential donors.
It is expected that the effect of the presence of several nested watersheds in a dataset is related to the effect due to
station density. Because of that, further purpose of the study is to compare the results obtained from the above described
nested catchments analysis to the impact of station density on regionalisation accuracy. Parajka et al. (2015) tested the
impact of the station density not for rainfall-runoff modelling but for the direct weighted interpolation of daily runoff
time-series with the topological-kriging (or Top-kriging) approach (see Skøien et al., 2006). Here, the same approach
for analysing the density is applied to all the parameters regionalisation techniques.
The full station density in the dataset is about 2.5 gauges per 1000 km$^2$, estimated dividing the total number of stations
by the area of Austrian territory, which is approximately 84000 km$^2$. All the applied regionalisation approaches are
tested for decreasing station density in the catchments dataset. Given a certain value of station density, the
corresponding number of gauged stations is randomly sampled from the original set of 209 catchments and the
regionalisation approaches are applied on this subsample (catchments input dataset) in leave-one-out cross validation: in


turn, each of the catchment in the subsample is considered to be ungauged and the remaining basins are used as
potential donors. Figure 5 shows an example of three samples for two different station densities, corresponding to 25
and 100 stations in the input dataset.
For each given value of station density, the following procedure is carried out:
-  100 different random samples (i.e. 100 different subsamples) with the same number of catchments are
generated.
-  for each subsample, the regionalisation approaches are applied, through leave-one-out cross validation and the
deterioration of the performances with decreasing density is analysed.

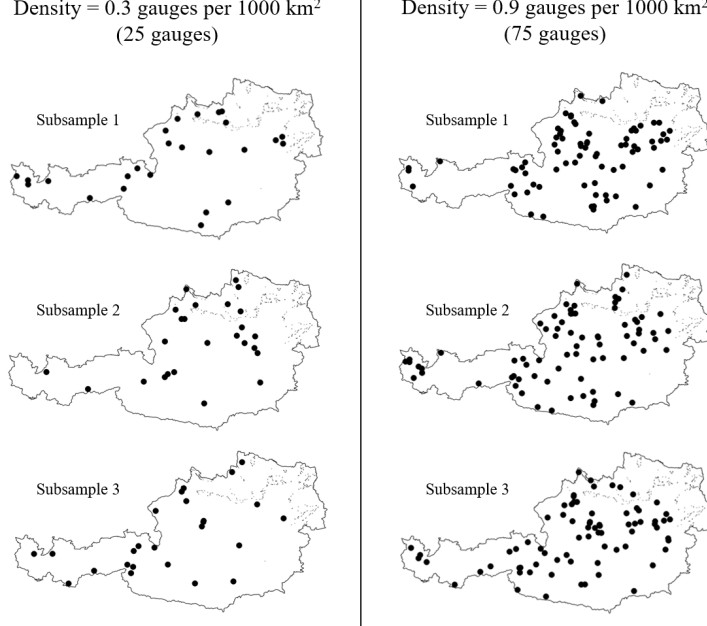


Figure 5. Example of three samples for two different station densities.

**4 Results and discussion**
**4.1 Model performances "at site"**
As anticipated, the rainfall-runoff models are calibrated against Kling-Gupta Efficiency (Eq. 1). In addition to KGE,
model performances are evaluated through Nash-Sutcliffe Efficiency (Eq. 8) as well. While KGE considers different
types of model errors (the error in the mean, the variability and the dynamics of runoff), NSE is a standardize version of
the mean square error.

$$NSE = 1 - \frac{\sum(Q_{sim} - Q_{obs})^2}{\sum(Q_{obs} - \overline{Q_{obs}})^2}$$  *Eq. 8*






where $Q_{sim}$ is the simulated runoff, $Q_{obs}$ is the observed runoff and $\overline{Q_{obs}}$ is the average observed runoff.
Table 4 shows the model performances obtained calibrating the models "at site", that is over the streamflow measured
in each catchment during the calibration period (1977-1992) and validated over the years 1992-2008 (no regionalisation
procedure is involved).
Both rainfall-runoff models behave well for the study area: in calibration, the median Kling-Gupta efficiencies are 0.85
for TUW and 0.88 for GR6J model, while in validation they deteriorate to 0.76 and 0.81 respectively. In the calibration
period, KGE is always above 0.66 and 0.76, respectively for TUW and GRJ6, whereas in validation, the KGE is over
0.72 for both models for 75% of the basins (even if it drops below 0.3 for two and one basins, respectively for TUW
and GR6J).
Looking at Nash-Sutcliffe efficiency the difference between the two models is even more marked than for the KGE:
GR6J model tends to perform better than TUW, despite the lower number of parameters.

Table 4. "At site" performances: values of the 25% (1st quar.), 50% (med.) and 75% (3rd quart.) quantiles for Kling-
Gupta (KGE) and Nash-Sutcliffe (NSE) efficiencies.

| | | KGE [-] | | | NSE [-] | | |
|---|---|---|---|---|---|---|---|
| | | 1st quart. | med. | 3rd quart. | 1st quart. | med. | 3rd quart. |
| **TUW** | **Calibration 1977 - 1992** | 0.82 | 0.85 | 0.90 | 0.65 | 0.72 | 0.80 |
| | **Validation 1992 - 2008** | 0.72 | 0.76 | 0.82 | 0.59 | 0.66 | 0.72 |
| **GR6J** | **Calibration 1977 - 1992** | 0.86 | 0.88 | 0.91 | 0.72 | 0.77 | 0.81 |
| | **Validation 1992 - 2008** | 0.75 | 0.81 | 0.84 | 0.67 | 0.74 | 0.79 |


### 388    4.2 Regionalisation performances using all catchments as potential donors

### 389    4.2.1 Choice of the donors for the "output averaging" regionalisation methods

Before comparing performances of regionalisation methods, it is necessary to choose the optimal settings for the output-
averaging versions of nearest neighbour (NN-OA) and "most similar" (MS-OA) techniques.
As anticipated in the methodology Section 3.2.5, we first investigate the effect of using different numbers of donors: in
particular, values between 1 and 50 are tested for both regionalisation techniques.
Regionalisation methods are repeated through leave-one-out cross-validation for each number of donors *n* and the
median Kling-Gupta efficiency obtained for each value of *n* over all the 209 catchments is computed. Tests are
performed for calibration and validation periods, but results are reported only for the validation period.
Figure 6 shows the median Kling-Gupta efficiency when the changing number of donors for TUW (upper panel) and
GR6J (lower panel). Looking at the figures, we may see that in all the four cases, the index always deteriorates when
more than 10 donors are chosen. On the other hand, there is not a unique optimal number of donors for the two models
nor for the two regionalisation techniques. The optimal number of donors identified according to the median of the
KGE varies between 3 and 7 depending both on the rainfall-runoff model (TUW or GRJ6) and on the regionalisation
approach (NN-OA or MS-OA). Since the KGE differences between 3 and 7 donors are not so relevant (around 0.02),





we decided to use 3 donors for both regionalisation methods and both models, which is also the most parsimonious
option. In addition, the choice of a low number of donors is convenient also in view of the analysis to be done on
decreasing density, where a large number of donors would imply the use of catchments that are less and less similar to
the target one.
It may be noted that the results by Oudin et al. (2008) highlighted a clearer pattern of model performances when
increasing the number of donors, with a stronger decrease in efficiency when using high numbers of donors. This may
be explained by the fact that they were using a simple not-weighted average of outputs. Here instead, the influence of
the additional donors is gradually poorer, due to the weights implemented in the output averaging procedure (Eq. 5):
when adding further donors to the approaches, the corresponding weights in the average are gradually lower according
to the increasing distance (for NN-OA) or dissimilarity index (for MS-OA) from the target. Thus, the impact of the less
similar catchments is smoothed, compared to what may be achieved using a not-weighted output average.

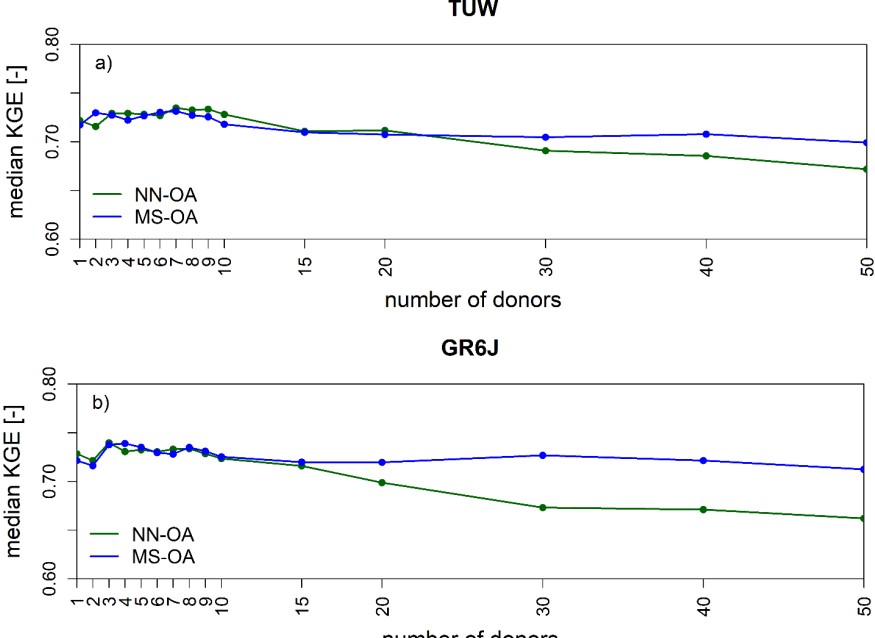

Figure 6. Impact of the number of donors on output-averaging nearest neighbour (NN-OA) and 'most similar' (MS-OA)
regionalisation methods for TUW (panel a) and GR6J (panel b) model.

**4.2.2 Performances of the regionalisation methods**
This section shows the performances of the regionalisation methods without excluding any candidate donor: the above
described regionalisation methods will be tested over all the 209 study catchments through leave-one-out cross
validation, for both models. Here all the basins in the dataset are used as potential donors: in turn, each basin is
considered to be ungauged and all the remaining (208) catchments are available in the donors set for testing the
regionalisation approaches. The parameter sets of the donor catchments used in the regionalisation are obtained through





a calibration procedure over the years 1977-1992, whereas for assessing the performances of the regionalisation
methods, only the results obtained over the validation period (1992-2008) are reported. Spatiotemporal transfer of
model parameters is therefore the most exacting task (as confirmed by the study of Patil et al. 2015), since we are using
parameters obtained over different catchment (in regionalisation) and over a different observation period. On the other
hand, this is exactly what would happen in a real-world forecasting application or for assessing the impact of a climate
change scenario, where you have to identify the parametrization of a model to be used for independent hydro-climatic
conditions and in any possible river section in the region.

Figure 7 reports Kling-Gupta and Nash-Sutcliffe efficiency boxplots for the two models when regionalising following
each of the techniques.
For TUW (Figure 7, panels a and b), all regionalisation methods provided good simulations: with respect to the
performances (always on the validation period) obtained when the models have been calibrated on the target section ("at
site" simulations, white boxes): the loss in efficiency indexes is, overall, limited. The Nash-Sutcliff efficiencies of KR,
MS-1 and NN-1 methods are consistent with the findings of Parajka et al. (2005), who computed only the NS: their
results are very similar to the present ones, even if they worked on a greater number of Austrian catchments and
calibrating the model against a different objective function.
For the GR6J model (Figure 7, panels c and d), the efficiencies of the nearest neighbours (NN-1 and NN-OA) and
"most similar" (MS-1 and MS-OA) regionalisations are closer to those of the TUW in respect to what happened when
the models are calibrated "at site". In fact, the GR6J model in regionalisation mode deteriorates more than HBV in
respect to the parametrization obtained considering the target as gauged.
In addition, we notice that, for this model, the ordinary kriging has performances always poorer than all the other
regionalisation methods.

For both rainfall-runoff models MS-OA tends to provide the best results and in general the two methods based on
"output average" (NN-OA and MS-OA), that exploit the information from more than one donor, outperform NN-1 and
MS-1, in particular in terms of Nash-Sutcliffe efficiency. This confirms the usefulness of regionalising on the basis of
more than one donor, as indicated by previous studies (e.g. McIntyre et al. 2005, Oudin et al. 2008, Viviroli et al. 2009,
Zelelew and Alfredsen 2014).

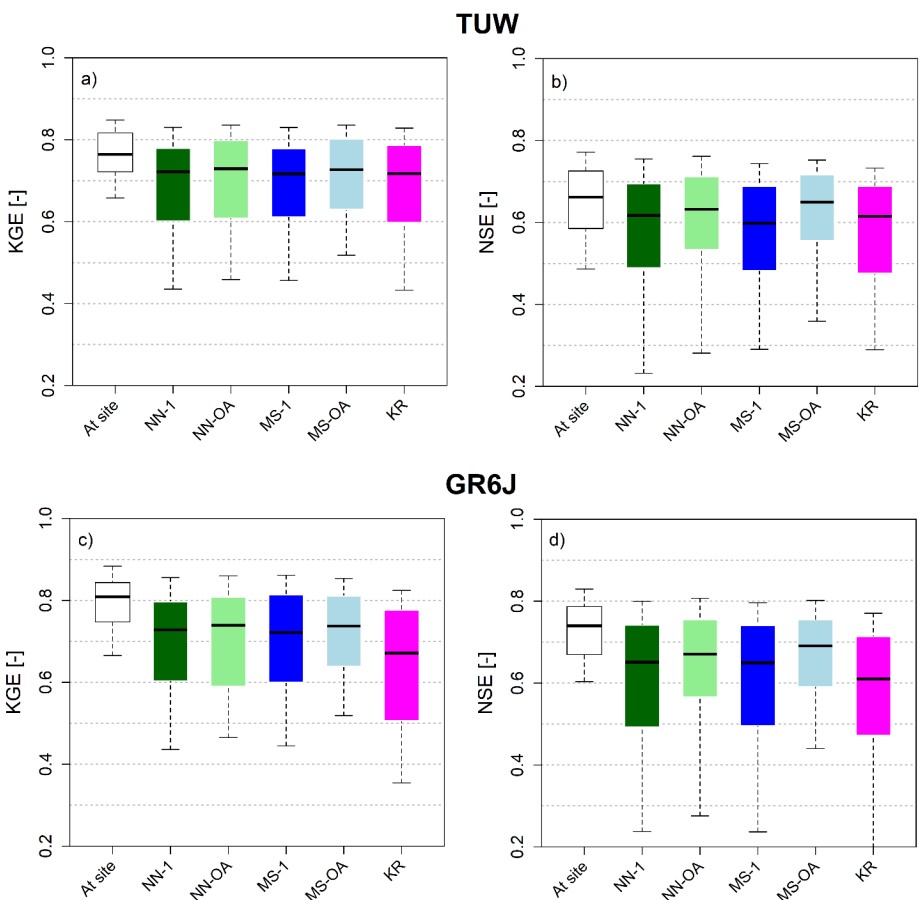

Figure 7. Original performances of the regionalisation methods for TUW model (panels a and b) and for GR6J model
(panels c and d) for the 209 Austrian catchments in the validation period 1992-2008. Boxes extend to 25% and 75%
quantiles while whiskers refer to 10% and 90% quantiles.

**4.3 Impact of nested donors: performance losses in regionalisation**
**4.3.1 Catchments identified as nested by the two criteria**
As introduced in Section 3.3, two different Criteria are implemented for identifying which donor catchments are
considered to be nested in relation to an ungauged catchment: *Criterion 1* (Figure 8, panel a) assumes that the only
nested donors are the first downstream and the first upstream gauged sections. Following this approach, 81% of the
catchments in the dataset have at least one downstream or upstream nested donor (red dots in Figure 9, panel a).
Instead, *Criterion 2* (Figure 8, panel b) excludes all the potential donors sharing a given percentage of drainage area
with the target catchment. It requires the definition of a percentage threshold value of shared drainage area. A
preliminary sensitivity analysis (not reported here) was performed, investigating the effect of different values between
5% and 20% for such percentage. Results show that differences in terms of regionalisation performance are not
significant and it is fixed to 10%. The choice of the threshold influences the number of catchments which can be
included in the study: in fact, the higher is the threshold, the lower is the number of basins classified as nested following





*Criterion 2*. Using 10% as a threshold allows to include most of the watersheds in the analysis: 65% (137 catchments)
of the basins have at least one nested donor catchment sharing at least the 10% of its area (red dots in Figure 9, panel b).
All the watersheds having potential nested donors according to the second criterion have nested gauged catchments also
according to the first criterion, but not vice versa: the impact of nested catchments on regionalisation performances is
therefore evaluated only for those 137 catchments which are considered to have nested gauged catchments following
both criteria.
It is important to highlight that the remaining 35% of the basins are still used as potential donor catchments, but the
regionalisation approaches are not repeated using such basins as targets (since they have no nested donors, their
performance would not change and they would distort the results).
Among the 137 catchments considered for the analysis of the "nestedness", 43% result to have only downstream nested
donor(s), 28% only upstream nested donor(s), and 29% at least one upstream and one downstream nested donors.

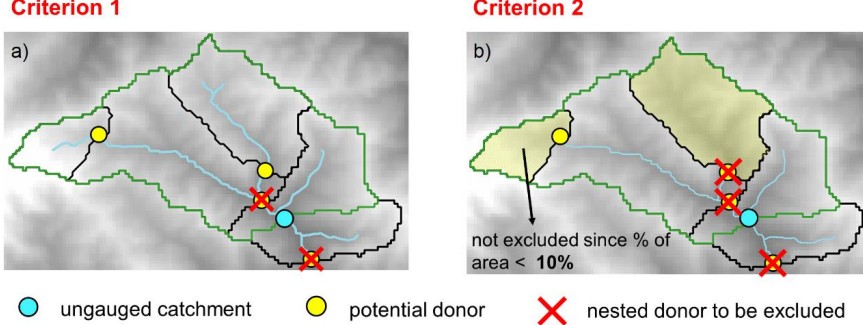


Figure 8. Criteria for excluding nested catchments when regionalising model parameters.





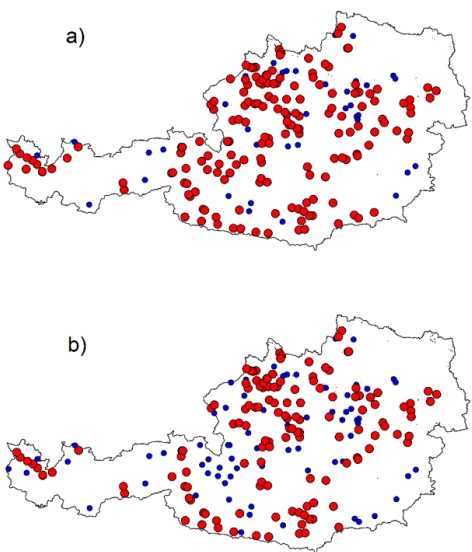


Figure 9. Panel a: red dots (170) refer to catchments with at least one upstream or downstream nested gauged
catchments (Criterion 1). Panel b: red dots (137) refer to catchments with at least one nested gauged catchment sharing
more than 10% of drainage area (Criterion 2).


### 490     4.3.2 Performance losses in regionalisation when excluding nested donors

The regionalisation methods are applied again in leave-one-out cross validation, this time excluding from the available
donors the catchments which are nested in relation to the target (ungauged) basin. This is done for both "nestedness"
criteria (down/upstream or overlapping of drainage area) and the analysis applies exclusively to the 137 catchments
classified as nested according to both criteria (red dots in Figure 9, panel b). The figures of this section (Figures 10 to
13) therefore refer to such subset.

Figures 10 and 11 compare the different performances (Kling-Gupta and Nash-Sutcliffe efficiencies in the upper and
lower panels respectively) obtained in regionalisation (always over the validation period), when nested catchments are
available or not as candidate donor basins for both TUW model (Figure 10) and GR6J (Figure 11). Each group of
boxplots refers to a different regionalisation method: within such groups, the first box indicates the performance when
no basins are excluded from the donor set, while the second and the third boxes report the performances due to the
exclusion of the nested following Criterion 1 or 2 respectively.

The performance deterioration is highlighted by bar plots in Figures 12 and 13, showing the mean loss in Kling-Gupta
and Nash-Sutcliffe efficiencies when excluding nested following the two criteria.

Finally, Table 5 reports the interquartile variability of Kling-Gupta e Nash-Sutcliffe efficiencies for both models and all
the regionalisation approaches when nested donors are excluded or not.






The method that is less affected is the ordinary kriging, especially for the HBV model, due to the fact that such method
is not based on the identification of one or more 'sibling' donors which may have been excluded if nested. On the other
hand, it should also be highlighted that such method is the regionalisation approach that performs worst, when nested
basins are available.

As expected, for both TUW and GR6J, NN-1 is always the most heavily affected method (dark green bars in bottom
panels of Figure 12 and 13): this is due to the fact that the nearest donor is a nested one in more than 80% of the
catchments, for both criteria and its exclusion seriously compromise the performance.

Excluding the nested catchments has also a strong impact on MS-1 (dark blue bars in bottom panels of Figures 12 and
13), even if to a lesser extent than for NN-1, since for more than 60% of the catchments the most similar donor is a
nested one according to both criteria.

The degradation of performance moving from Criterion 1 (upstream/downstream) to Criterion 2 (overlapping drainage
area) highlighted in Figure 10 and 11 demonstrates that using as donors not only the immediate downstream or
upstream gauged river sections, but also all the catchments partially sharing their drainage area with the target one, have
a strong positive influence on the regionalisation performance.

Furthermore, it is clear how the use of output-averaging for both nearest neighbour and "most similar" approaches (NN-
OA and MS-OA), in addition to perform better than the NN-1 and MS-1 when using all (nested and non-nested) donors
(see also Section 4.3) can also improve the robustness of the methods to the exclusion of the nested donors: the bottom
panels of Figures 12 and 13 in fact show that the loss in the efficiencies of  NN-OA and MS-OA are always smaller
than those corresponding to the single donor approaches (NN-1 and MS-1), for both rainfall-runoff models and for both
regionalisation methods. This confirms that the use of output-averaging (or more in general the use of more than one
donor basin) is preferable for regionalisation purposes also for regions that do not have so many nested catchments as
the Austria study area.

Finally, the values reported in Table 5 (as well as Figure 12 and 13) shows how, especially for NSE, the losses resulting
when excluding nested donors from the regionalisation are higher for the GR6J model than for the HBV: the GR6J
seems to be slightly more affected by the presence of nested basins, except for MS-1 and MS-OA whose performances
remain more similar to those of TUW.



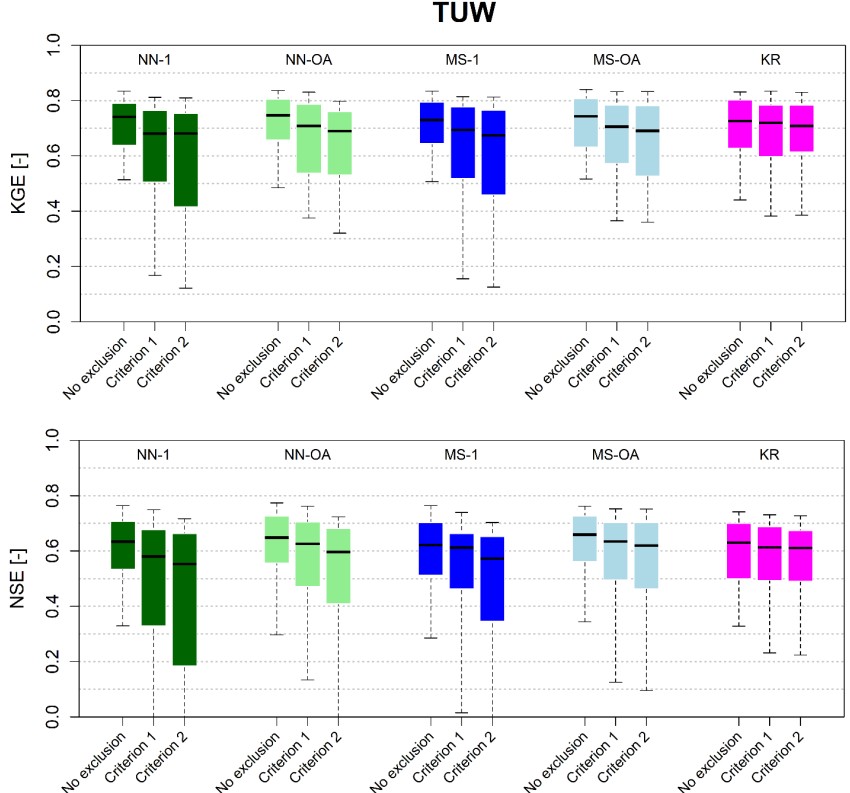

Figure 10. Effect of the exclusion of nested catchments for the subset of 137 watersheds classified as nested: Kling-
Gupta (upper panel) and Nash-Sutcliffe (lower panel) efficiencies when regionalising the TUW model. "No exclusion":
all the donors are available. "Criterion 1" or "Criterion 2": nested catchments are excluded from donor set. Box colours
refer to the different methods: green is nearest neighbour (1 donor is dark green and 3 is light green), blue is most
similar (1 donor is dark blue and 3 is light blue) and magenta is ordinary kriging. Boxes extend to 25% and 75%
quantiles while whiskers refer to 10% and 90% quantiles.






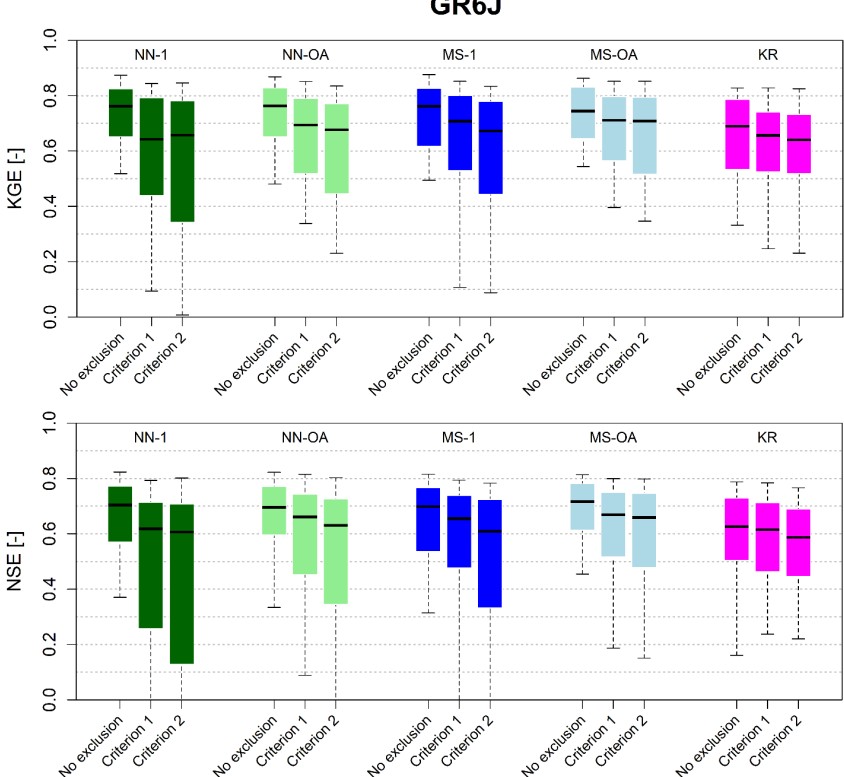

Figure 11. Effect of the exclusion of nested catchments for the subset of 137 watersheds classified as nested: Kling-
Gupta (upper panel) and Nash-Sutcliffe (lower panel) efficiencies when regionalising the GR6J model. "No exclusion":
all the donors are available. "Criterion 1" or "Criterion 2": nested catchments are excluded from the donor set. Box
colours refer to the different methods: green is nearest neighbour (1 donor is dark green and 3 is light green), blue is
most similar (1 donor is dark blue and 3 is light blue) and magenta is ordinary kriging. Boxes extend to 25% and 75%
quantiles while whiskers refer to 10% and 90% quantiles.





**TUW**

No nested excluded

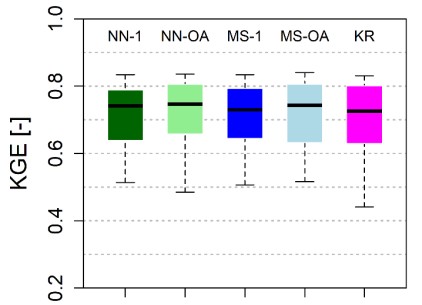
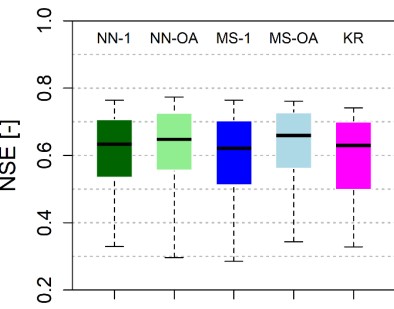

Loss in regionalisation performances
when excluding nested donors

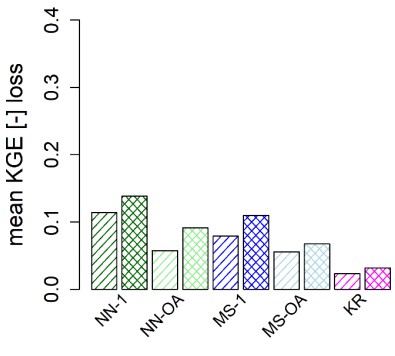
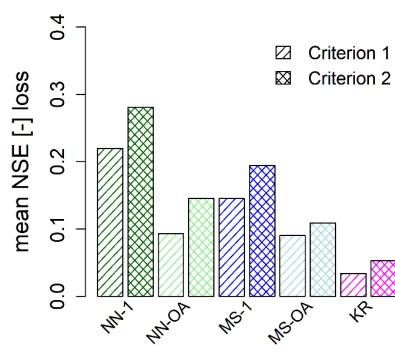

Figure 12. Kling-Gupta and Nash-Sutcliffe efficiencies and mean losses in the same criteria resulting when excluding
the nested donors with Criterion 1 and 2 (bottom panels) for TUW model.





## GR6J

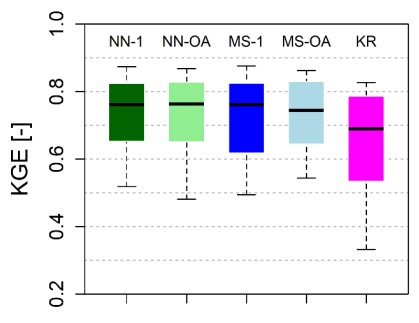

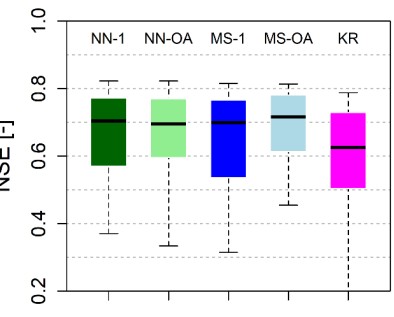

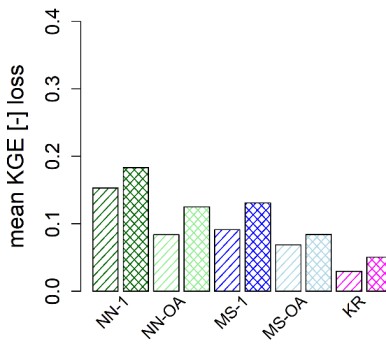

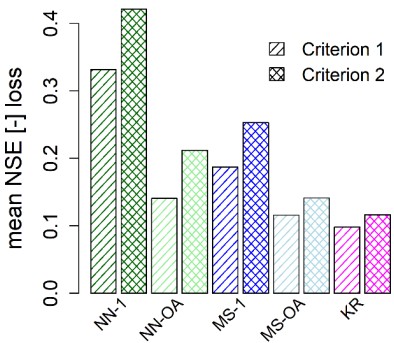

Figure 13. Kling-Gupta and Nash-Sutcliffe efficiencies and mean losses in the same criteria resulting when excluding the nested donors with Criterion 1 and 2 (bottom panels) for GR6J model.

Table 5. Inter-quartile values of Kling-Gupta and Nash-Sutcliffe efficiencies when regionalising TUW and GR6J models excluding or not excluding nested donor catchments.

| | | **Inter-quartile KGE [-]** | | | | |
|---|---|---|---|---|---|---|
| | | **NN-1** | **NN-OA** | **MS-1** | **MS-OA** | **KR** |
| **TUW** | **No nested excluded** | 0.64/0.79 | 0.66/0.81 | 0.64/0.79 | 0.63/0.81 | 0.63/0.80 |
| | **Criterion 1** | 0.50/0.76 | 0.54/0.79 | 0.52/0.78 | 0.57/0.78 | 0.60/0.78 |
| | **Criterion 2** | 0.42/0.75 | 0.53/0.76 | 0.46/0.77 | 0.53/0.78 | 0.61/0.78 |
| **GR6J** | **No nested excluded** | 0.65/0.82 | 0.65/0.83 | 0.62/0.83 | 0.64/0.83 | 0.53/0.79 |
| | **Criterion 1** | 0.44/0.79 | 0.52/0.79 | 0.53/0.80 | 0.56/0.80 | 0.52/0.74 |
| | **Criterion 2** | 0.34/0.78 | 0.45/0.77 | 0.44/0.78 | 0.52/0.79 | 0.52/0.73 |
| | | **Inter-quartile NSE [-]** | | | | |
| | | **NN-1** | **NN-OA** | **MS-1** | **MS-OA** | **KR** |
| **TUW** | **No nested excluded** | 0.53/0.71 | 0.56/0.73 | 0.51/0.70 | 0.56/0.73 | 0.50/0.70 |
| | **Criterion 1** | 0.33/0.68 | 0.47/0.70 | 0.46/0.66 | 0.50/0.70 | 0.49/0.69 |
| | **Criterion 2** | 0.18/0.66 | 0.41/0.68 | 0.35/0.65 | 0.46/0.70 | 0.49/0.67 |
| **GR6J** | **No nested excluded** | 0.57/0.77 | 0.60/0.77 | 0.54/0.77 | 0.61/0.78 | 0.50/0.73 |
| | **Criterion 1** | 0.26/0.71 | 0.45/0.74 | 0.48/0.74 | 0.52/0.75 | 0.46/0.71 |
| | **Criterion 2** | 0.13/0.71 | 0.34/0.73 | 0.33/0.72 | 0.48/0.75 | 0.45/0.69 |





**4.4 Impact of station density: performance losses in regionalisation**

The last results concern the analysis on the impact of station density on regionalisation performances. As introduced in Section 3.4, for decreasing values of station density across Austria, 100 random samples of stream gauges are generated from the 209 catchments data set and the regionalisation methods are repeated in leave-one-out cross validation over each one of such samples. Seven different values of station density from 0.3 to 2.1 gauges per 1000 km$^2$ are tested, which correspond to a total number of stations across Austria from 25 to 175.

For each assigned density value, the described procedure provides 100 different sets of regionalised target catchments. For a given density, each one of these 100 subsamples is formed by the same number of target catchments, resulting therefore in the same number of efficiencies to be analysed.

In order to analyse the results, the median regionalisation performances of each subsample are computed and presented here: thus, for each gauging density, the results consist in 100 values of median performances.

For sake of brevity, only the median Kling-Gupta efficiencies over the validation periods are reported. They are shown in Figure 14 for TUW model and in Figure 15 for GR6J model: each plot contains the boxplots of the median Kling-Gupta efficiencies for each station density (i.e. number of gauges per 1000 km$^2$), that is, each boxplot presents the 100 values of median Kling-Gupta efficiencies obtained applying the regionalisation approaches to the 100 subsamples generated with an assigned density. The colored point and the dotted line in the plots indicate the "original" (and maximum) median regionalisation efficiency of the approaches, that is the one obtained when using all available donors (i.e. actual station density, corresponding to 2.4 gauges/1000 km$^2$).

The NN-1 method (Figures 14 and 15, panels a) is the most affected by the decreasing density. In fact, when the density declines, there is an higher probability that the less dense subsamples do not include the catchment that is the nearest one to each target river section. And, as we have seen in the analyses on the nested donors, in the large majority of the cases, the nearest catchment is a nested one, whereas the second best may be substantially different from the target basin.

Also the output-averaging version of the nearest neighbours method (Figures 14 and 15, panels b) strongly deteriorates for less dense networks. In general, nearest neighbour methods are highly sensitive to gauging density: geographical distance results to be a good similarity measure only for densely gauged study area (like Austria), since they firmly rely on the presence of gauged catchments in the immediate surroundings that are also hydrologically very similar. If the density decreases, the closest donor may be relatively far from the target, and it may therefore have little in common with it.

As far as the MS-1 (Figures 14 and 15, panels c) is concerned, its performances degrades more gracefully (with the exception of the GR6J model for the minimum density) than the NN-1 or the NN-OA. Also in this case (like for the NN-1), when the density decreases it becomes less probable that the most hydrologically similar catchment (identified by MS-1 in full density) is still part of the subsample; but it is also true there is more than one catchment in the original data set that is similar enough to the target in terms of catchment attributes.

This holds also for the output–averaging MS (Figures 14 and 15, panels d), which is even less affected by a reduction in donors' density and is the best-performing approach for any density (for both rainfall-runoff models).

We may note that, also in this analysis, analogously to what resulted for the exclusion of nested catchments, for both approaches (NN and MS), the implementation of output-averaging allows to reduce the degradation in the performances in comparison to the corresponding 1-donor version.





The impact of station density is similar to that of excluding nested catchments also for the ordinary kriging approach
(Figures 14 and 15, panels e), which deteriorates less than the other methods for decreasing values of station density.
For the TUW model, the kriging regionalisation, starting from an already high KGE in full density, results in
performances that are inferior only to those of MS-OA when the density goes below 0.9. For the GR6J model, even if
the deterioration is limited, since the kriging was poorly performing for the full density regionalisation (Figure 7), the
median KGE is always worse than those of all the other regionalisation approaches, for all the station densities.
Overall, all methods (excluding the poorly performing NN-1 and the kriging for the GR6J) result in relatively good
performances provided that the station density is at least 0.9 gauges per 1000 km$^2$. On the other hand, leaving aside the
kriging method, the median KGE drops very steeply when the density passes from 0.6 to 0.3 gauges per 1000 km$^2$.

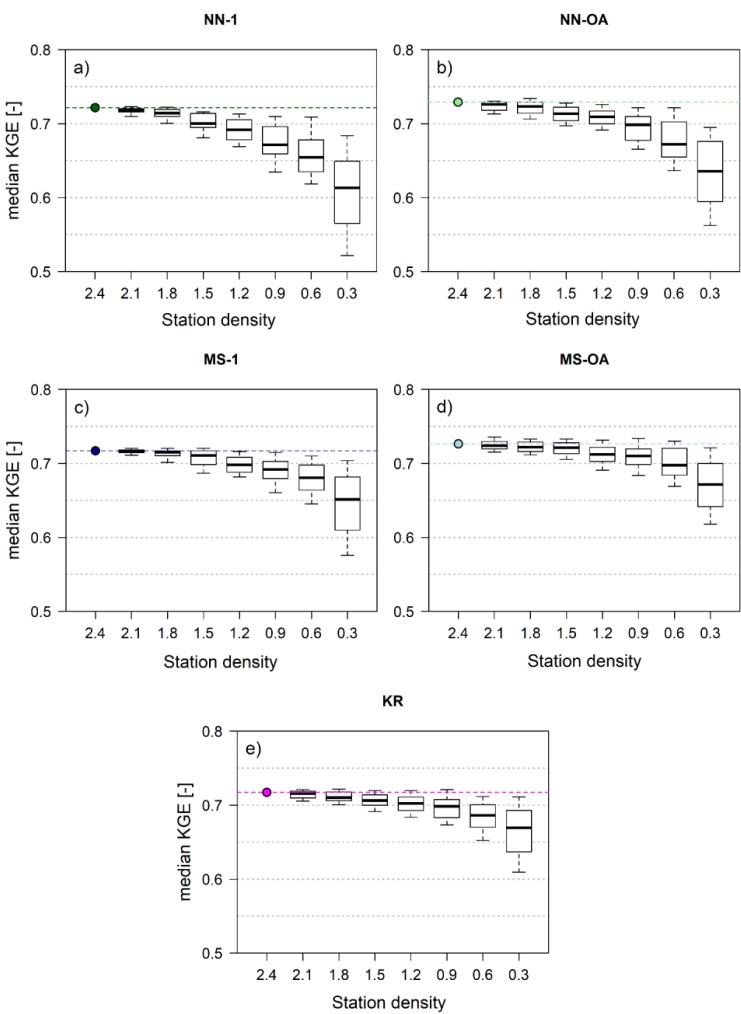

Figure 14. Median Kling-Gupta efficiency of the 100 sampled datasets for varying station density (number of gauges
per 1000 km$^2$) for the TUW model using NN-1 (panel a), NN-OA (panel b), MS-1 (panel c), MS-OA (panel d) and KR
(panel e) regionalisation methods. The colored point and dotted line in the plots indicate the original median
regionalisation efficiency of the approaches when using all available donors (i.e. actual station density, corresponding to
2.4 gauges/1000 km$^2$).

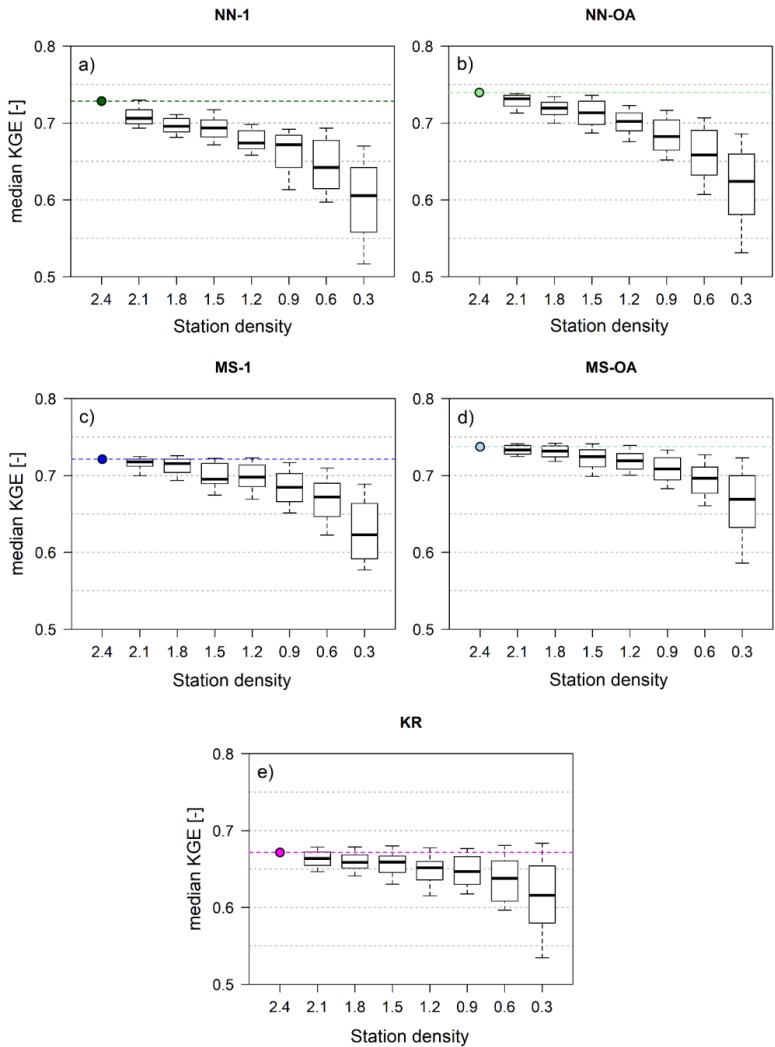

Figure 15. Median Kling-Gupta efficiency of the 100 sampled datasets for varying station density (number of gauges
per 1000 km$^2$) for the GR6J model using NN-1 (panel a), NN-OA (panel b), MS-1 (panel c), MS-OA (panel d) and KR
(panel e) regionalisation methods. The colored point and dotted line in the plots indicate the original median
regionalisation efficiency of the approaches when using all available donors (i.e. actual station density, corresponding to
2.4 gauges/1000 km$^2$).

**5 Discussion and conclusions**
An assessment of the impact of the presence of nested catchments and of station density on the performance of
parameter regionalisation techniques in a large Austrian dataset has been performed. The main motivation for this work
lies in the lack of systematic studies in the literature about the effect of data-richness and information content when
evaluating the accuracy of various methods for transferring rainfall-runoff model parameters to ungauged catchments.
In fact, studies conducted on different study sets often do not lead to the same ranking of the tested approaches and the



obtained results are not extendable to different study regions. This is indeed due also to the diverse topological
relationships between catchments ("netstedness") in the datasets and to the diverse density of the streamgauges.
The purpose of the work is to give support to the choice of the most appropriate parameter regionalisation approaches,
showing and quantifying if and how the presence of several nested catchments in a dataset or the gauging density can
distort the predictive power of a certain technique.
The research has been conducted for a very densely gauged dataset covering a large portion of the Austrian country.
Two rainfall-runoff models for simulating daily streamflow have been calibrated for the 209 study watersheds: a semi-
distributed version of the HBV model (TUW model), and the lumped GR6J model coupled with the Cemaneige snow
routine.
Both models perform very well when applied in "at-site" mode, that is when parameterised in the traditional, (not
regionalised) way, and for each target section the historical gauged streamflow data are used for fitting the parameter
set. The calibration and validation performances are very good for both rainfall-runoff models, with better values of the
chosen goodness-of-fit indexes for the GR6J model, which demonstrates to perform very well also in this Alpine
dataset.
In order to assess the capability of the models when used on ungauged basins, the streamgauge data for every section
was, in turn, considered not to be available, and five regionalisation approaches were implemented for using the
rainfall-runoff models in such 'ungauged' sections over the validation period. This is indeed an exacting task because
we are attempting to use the model over an ungauged catchment and for an observation period different from the one
used for parameterising the gauged donor catchments. The first regionalisation approach is an ordinary kriging
approach (KR), which separately interpolates each of the model parameter based on their spatial correlation in the study
area. Two approaches selecting one single donor catchment and transposing its parameter set to the target basin are also
tested: in the first (NN-1) the geographically nearest catchment is selected, while in the second approach (MS-1) the
single donor that "lends" all its parameters to the target one is the most similar one in terms of a set of physiographic
and climatic attributes. The latter two approaches are implemented also in the output-averaging (OA) version, where the
entire parameter set of more than one donor is used for the simulation on the target section and the model outputs are
then averaged accordingly to the distance/dissimilarity between donors and target.
In regionalisation mode, the performances of the GR6J model deteriorates more than those of the TUW model, in
comparison with the 'gauged', at-site parameterisation. For both rainfall-runoff models, the use of the output averaging
approach outperform the use of a single donor (NN-OA and MS-OA performed better than NN-1 and MS-1),
confirming the outcomes of other studies on the importance of exploiting the information available from more than only
one donor (see e.g., McIntyre et al. 2005, Oudin et al. 2008, Viviroli et al. 2009, Zelelew and Alfredsen 2014). The
output-averaging methods also outperform the parameter-averaging kriging method (especially for the GR6J model),
showing that it is preferable transferring the entire parameter set of each donor, thus maintaining the correlation
between the parameter values. The results of the MS-OA are close but tend to be better than those of the NN-OA,
indicating that hydrological similarity is more important than geographical closeness for choosing the donors.
We expect that spatial proximity alone may be even less representative of hydrological similarity in a drier climate: in
fact Patil et al. (2012) and Li and Zhang (2017) shown that in dry runoff-dominated regions, nearby catchments tend to
exhibit less hydrological similarity than in more humid regions.




The impact of the "richness" of the data set was then analysed, in order to assess the deterioration of the regionalisation
approaches for decreasing availability and 'worth' of the available donors, starting from the influence of using nested
basins as donors.

Two criteria have been proposed for identifying a basin that is nested with the target one: the first one, already used in
the few analysis of "nestedness" in the literature, classifies as nested the first upstream and the first downstream gauges
on the river network. The second, novel criterion, identifies as nested all the catchments that share more than a given
percentage (here chosen as 10%) of the drainage area with the target one. It results that the first criteria includes in the
list of basins being nested with at least one potential donor many more sections than those identified by the second
criteria. In fact, the first criterion considers as nested also a number of catchments that share less than 10% of area with
the target one: this means that, in some cases, the first downstream or upstream gauge may be not representative of the
same drainage area and their catchments may be governed by very different hydrological processes.

All the regionalisation approaches have been repeated by excluding from the donor set the catchments assumed to be
nested in relation to each target basin, according to each one of the two criteria.
For both rainfall-runoff models and for all the regionalisation approaches, when using the second criterion (that
excludes all the basins that share a significant portion of the same watershed), the regionalisation procedure deteriorates
more than when excluding the first up/downstream river sections, whose catchment may, in some cases, not have much
in common with the target one.
Looking at the two rainfall-models, when excluding the nested catchments, the regionalisation performances tend to
deteriorates more for the GR6J than for the TUW: this seems to indicate that the TUW model may be more robust for
regionalisation purposes, even when nested donors are not available.
Comparing the different regionalisation approaches, the parameter-averaging kriging is the method that is less impacted
by the exclusion of the nested donors, since it does not depend only on the choice of one or few 'sibling' donors, that
are very often the nested ones, but it takes into account a number of donors in a given radius. This is consistent to the
outcomes of Merz and Blöschl (2004) and Parajka et al. (2005) who observed almost no deterioration of regionalisation
performances when excluding the first down and upstream nested donors using the same ordinary kriging approach.
When using, instead, a method transferring the entire parameter set from one or more donor catchments, the
deterioration is more sizeable. The method that experiences the worst deterioration is the NN-1, since in 80% of the
cases, the nearest basin is a nested one, and it is thus excluded from the potential donors; second worst is the MS-1, that,
when free to choose any single potential donor in the entire region, would choose a nested one in 60% of the cases. The
output-averaging methods degrade less severely, showing that exploiting the information resulting from more than one
donor increases the robustness of the approach also in regions that do not have so many nested catchments as the
Austrian one (where the importance of nested donors in regionalising model parameters is highlighted also by Merz and
Blöschl, 2004).

Finally, an assessment of the impact of station density on the regionalisation has been also implemented. The nearest
neighbour approaches (both NN-1 and NN-OA) are the methods that suffer more from the decrease in gauging density,
whereas the "most similar" methods (MS-1 and MS-OA), which use as similarity measure a set of catchment
descriptors, are more capable to adapt to less dense datasets: in fact the "most similar" methods are able to find other





adequate donors, that may be anywhere in the region, whereas the nearest neighbours techniques, in a more 'sparse'
monitoring network risk to identify a "not so near" donor that may be very different from the target one.
The impact of decreasing station density on the performance of the output-averaging approach based on spatial
proximity (NN-OA) is in line to what observed by Lebecherel et al. (2016).
The performances of both the output-averaging methods, in agreement with the results obtained for similar methods by
Oudin et al. (2008), strongly deteriorate when the station density drops below 0.6 gauges per 1000 km$^2$.

The study confirms how the predictive accuracy of parameter regionalisation techniques strongly depends on the
information content of the dataset of available donor catchments, quantifying the contribution of nested catchments and
station density for different approaches and rainfall-runoff models. The outcomes obtained in reference to the Austrian
data set indicate that the reliability and robustness of the regionalisation of rainfall-runoff model parameters can be
improved by making use of output-averaging approaches, that use more than one donor basin but preserving the
correlation structure of the parameter set. Such approaches result to be preferable for regionalisation purposes in both
data-poor and data-rich regions, as demonstrated by the analyses on the degradation of the performances resulting from
either removing the nested donor catchments or decreasing the gauging station density.

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

**Appendix A: Choice of best catchment descriptors**
The implementation of the "most similar" approach requires the choice of the geo-morphologic and climatic attributes
to be used for selecting the donor catchment(s), i.e. to calculate the dissimilarity indices of equation 7.
This similarity study is part of a preliminary analysis carried out using the whole period of available daily data (from
1976 to 2008, again with 1 year of warm-up) for calibrating the rainfall-runoff models.
In order to individuate the best catchment descriptors (all reported in Table 1 with a brief description), the most similar
approach with one single donor catchment (MS-1) is applied sequentially to the entire dataset in leave-one-out cross-
validation, using at each step an increasing number of attributes when defining the dissimilarity index $\phi$. At each step,
the method is tested multiple times, adding one by one each of the attributes and the one which gives the best
regionalisation performances is selected. For greater clarity, Figure A1 (panel a) refers to TUW and panel b) to GR6J)
shows the boxplots of the consecutive best combinations of descriptors: at the first step, only one attribute is used, the
most similar approach is tested for all the available catchment features, and the similarity in the land cover classes
(Corine) gave the best efficiency. At the second step, the operation is repeated using land cover and each of the
remaining attributes one at a time, finding the geology classes to be the best attribute to add, and so on. The analysis
stops when the performances are decreasing or stop improving.



As can be inferred from Figure A1, both rainfall-runoff models reach good regionalisation performances when using up
to 5 attributes. Since the first best 5 attributes are the same for both models and from the sixth step the performances are
not substantially improved, we decide to choose those five descriptors to characterize catchment similarity: land use
classes, geological classes, mean annual precipitation, stream network density and mean elevation.

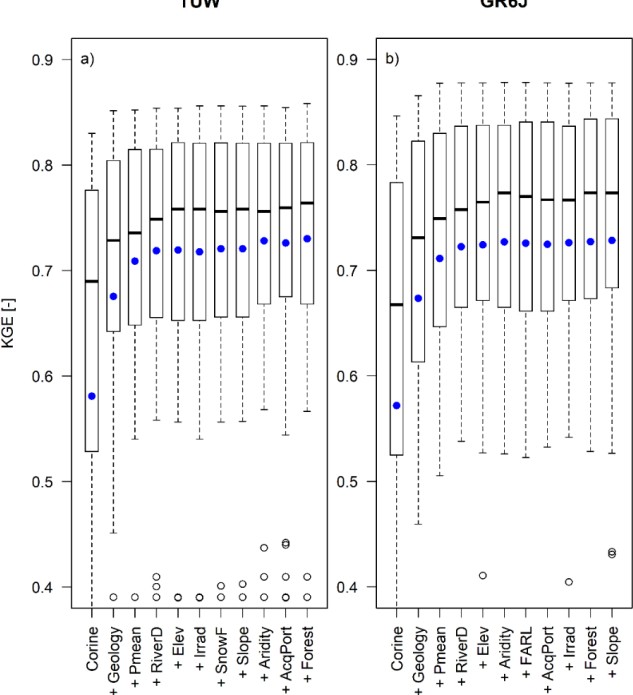


Figure A1. Kling-Gupta efficiencies for TUW (a) and GR6J (b) models for the consecutive steps of the similarity
analysis. Boxes refer to 25% and 75% quantiles, whiskers refer to 10% and 90% quantiles and the blue points to the
average.