# Peer review of "Importance of the informative content in the study area when regionalising rainfall-runoff"

_Hydrology and Earth System Sciences, 2020_

## Referee Comment (RC1) · Anonymous Referee #1 · 18 Apr 2020

This paper compared several kinds of regionalization strategies with two kinds of rainfall-runoff models. Generally speaking, this paper is well written and provides convincing evidence to support the conclusion. The title, abstract, introduction and discussion should be substantially revised. Here are some detailed comments.

1. The title is inconsistent with the numerical experiments of this paper. In this paper, 5 kinds of regionalization approaches (KR, NN-1, MS-1, NN-OA, MS-OA) were comprehensively compared with test data from Austria. The effects of nested catchments and gauging station density were also evaluated. Two rainfall-runoff models (TUW and

[Figure]

GR6J) were used in the evaluation and similar results can be obtained from different models, indicating the conclusion of this work can potentially be extended to various models. However, the key words 'Importance of the information content' has not been intensively discussed in the discussion part (please search 'information'). The concept 'information content' has not been well defined in this paper, and it seems not very necessary to have it defined. This paper is more likely to be a 'inter-comparison' paper instead of a 'concept promoting' paper. Consequently, I suggest the authors to revised the title in order to sharply and concisely show what have been done in this paper in a few words.

2. The abstract is too long! Please briefly introduce your work in a single paragraph. A wordy 'abstract' is not friendly to your potential readers. It is uncomfortable to locate useful information from a 'one-page, 5 paragraphs abstract'.

3. line 43-53: In order to clarify the relationships between different regionalization approaches, please add a 'mind map' about: regression-based, distance-based, output-averaging, parameter-averaging.

4. line 67: the best the best?

5. line 155: routing routine?

6. Discussion and conclusion part.

In this paper the hydrographs of simulated and observed streamflow were not presented. Although this is not necessary, a hydrograph may possibly intuitively show a lot of detailed mismatching patterns which are helpful for model diagnose. I suggest to add hydrographs from a typical catchment to compare not only the KGE and NSE criterions but also the details of systematic errors of different regionalization approaches. KGE and NSE criterions are good objective functions for optimization, but hydrographs can tell more useful information about the physical processes.

Consequently, the discussion part of this paper is mainly about explaining the results,

but lack of physical reasons. This paper has too much statistics but not enough hydrological mechanics on explaining the performance of different regionalization approaches. A comparison of regionalized parameters is strongly suggested to be added in the discussion part in order to provide fundamental physical reasoning of regionalization. For an example, please compare the parameter values from KR, NN and MS methods in a typical catchment, and discuss the influence of parameter values to corresponding hydrographs.

---

## Referee Comment (RC2) · Anonymous Referee #2 · 29 Apr 2020

The authors present a very good work on assessing the reliance of model regionalization approaches on the gauged data content. The methodology is clear and easy to follow. Their results indicate that transferring the entire parameter set, thus keeping the correlation among model parameters, outperforms the parameter-averaging kriging method. The output-averaging methods using more than one donor basins is more robust than the one-donor method. The most similar method based on geomorphological and climatic descriptors tends to show higher transferability in sparsely gauged areas than the nearest neighbor method. The findings provide important ref-

erence for the community when conducting hydrological modeling in sparsely or un-gauged basins. My major concern is that the structure of the manuscript is a bit scattered. The abstract and results sections are lengthy, from my taste. The authors could consider potential rephrase or reorganization. In addition, some Paragraphs in the results section should move to the methodology section. My second concern is that the applied two hydrological models have different calibrated/regionalized parameter spaces. In particular, the TUW model implies 11 parameters for calibration, while the GR6J model has less 8 parameters for calibration/regionalization. Some discussions on the effects of the size of parameter space on the results are needed. Third, the studied 209 catchments vary across a large range of area from 13 to 6000 km2. But, the models didn't use distributed parameter values. That is, the spatial variability of parameters in large basins wasn't considered. The author should also add discussions on the performance of the regionalization approaches in catchments with distinct basin areas (small, moderate and large basin). Finally, the authors should also specify how they sampled catchment subsets from the total 209 catchments when investigating the effects of station density. Did they sample manually or using automatic scripts? The thing here is how to guarantee that the sampled catchments are evenly distributed across the country. The sampled catchments concentrating in a small region could result in a same station density with the evenly distributed catchments. Some specified comments: 1. Please, keep "HBV" and "TUW" consistent. Otherwise, the readers would find three hydrological models in this study. Actually, there are only tow models; 2. Lines 8-10, consider to move to introduction section; 3. Lines 21-22, may consider to remove; 4. Lines 33-34, you already specified "how" above. Consider to remove; 5. Line 67, "the best the best"; 6. Line 96, what do you mean by "continuous simulating daily models"? 7. Line 116, "the more"; 8. Line 259, how to calculate "stream network density", please specify this; 9. Lines 305-322, consider to move that to introduction. Here, you could add description on your method to choose the number of donor catchments; 10. Line 289, by running; 11. Lines 366-373, move to methodology section; 12. Line 392, "as anticipated"—"as introduced"; 13. Line 402, so relevant—so obvious;

14. Lines 425-426, 460-471, move to methodology; 15. Line 507, KGE and NSE; 16. Lines 528-533, consider to rephrase that. Hard to follow; 17. Lines 538-540, could you add some discussions on that? 18. Lines 570-577, consider to move to methodology; 19. Lines 685-687, please rephrase that, cannot follow; 20. Please, consider to reduce the number of figures. Try to aggregate Figures 1 and 2. One of the options is try to present the performance of the two hydrological models in one figure instead of showing separately, such as figures 10-11, 12-13, 14-15. Or you may consider to provide the figures in a supplementary file.

---

## Author Response (AR1)

**Response to Reviewers**

The first part of the document reports the responses to the two anonymous referees, indicating also the relevant changes in the manuscript (the lines in the replies refer to the marked document copied in the following). The second part is a marked-up version of the revised manuscript (main changes and new parts are in blue font).

**Response to Reviewer 1**

**General comments:**
This paper compared several kinds of regionalization strategies with two kinds of rainfall-runoff models. Generally speaking, this paper is well written and provides convincing evidence to support the conclusion. The title, abstract, introduction and discussion should be substantially revised. Here are some detailed comments.

**Specific Comments:**
**1.** The title is inconsistent with the numerical experiments of this paper. In this paper, 5 kinds of regionalization approaches (KR, NN-1, MS-1, NN-OA, MS-OA) were comprehensively compared with test data from Austria. The effects of nested catchments and gauging station density were also evaluated. Two rainfall-runoff models (TUW and GR6J) were used in the evaluation and similar results can be obtained from different models, indicating the conclusion of this work can potentially be extended to various models. However, the key words 'Importance of the information content' has not been intensively discussed in the discussion part (please search 'information'). The concept 'information content' has not been well defined in this paper, and it seems not very necessary to have it defined. This paper is more likely to be a 'inter-comparison' paper instead of a 'concept promoting' paper. Consequently, I suggest the authors to revise the title in order to sharply and concisely show what have been done in this paper in a few words.
**Reply:** We thank the referee for suggesting more clarity in highlighting what the paper presents: the main focus and novelty of the work is the analysis of the role of the available data set, that is which and how many gauged catchments are available (to be used as donors) for the regionalisation (this is what we mean with information content of the data set: we will explain it better). And in order to do so, we compare different regionalisation approaches, to understand which approaches are more or less impacted by a change in the donors' data set (and, as you wrote, we applied two well-known rainfall-runoff models, for generalisation purposes). We agree with the referee that the term "information content" may confuse the reader, since it could be also associated to the Shannon's definition in the 'information theory' discipline, for the hydrological literature in this period.
In order to further clarify our main focus, we have:
  i)     modified the introduction (lines 56-57 and 92-96) and the conclusions (lines 637-640 and 680)
  ii)    replaced the term "information content" with "informative content" in both title and manuscript, which keeps the meaning we want to give but it avoids referring to other research subjects.

**2.** The abstract is too long! Please briefly introduce your work in a single paragraph. A wordy 'abstract' is not friendly to your potential readers. It is uncomfortable to locate useful information from a 'one-page, 5 paragraphs abstract'.
**Reply:** As suggested, we have substantially cut and revised the abstract, in order to have a more useful and agile summary (and also here focussing more on our main objective).

**3.** line 43-53: In order to clarify the relationships between different regionalization approaches, please add a 'mind map' about: regression-based, distance-based, output-averaging, parameter-averaging.
**Reply:** We have tried to clarify it, reorganising the sentence in bullet points (so that is more 'graphical') in order to make the difference between the approaches easier to understand (lines 35-47).

**4.** line 67: the best the best?
**Reply:** We have corrected this typo.

**5.** line 155: routing routine?

**Reply:** Yes, it is the programming routine used for computing the flow response and routing. It is the same terminology used in the TUW model manual but we agree that it is an alliteration and it does not "sound" well, and we have replaced "routine" with "module" (lines 151-152).

**6.** Discussion and conclusion part.

In this paper the hydrographs of simulated and observed streamflow were not presented. Although this is not necessary, a hydrograph may possibly intuitively show a lot of detailed mismatching patterns which are helpful for model diagnose. I suggest to add hydrographs from a typical catchment to compare not only the KGE and NSE criterions but also the details of systematic errors of different regionalization approaches. KGE and NSE criterions are good objective functions for optimization, but hydrographs can tell more useful information about the physical processes.

Consequently, the discussion part of this paper is mainly about explaining the results, but lack of physical reasons. This paper has too much statistics but not enough hydrological mechanics on explaining the performance of different regionalization approaches. A comparison of regionalized parameters is strongly suggested to be added in the discussion part in order to provide fundamental physical reasoning of regionalization. For an example, please compare the parameter values from KR, NN and MS methods in a typical catchment, and discuss the influence of parameter values to corresponding hydrographs.

**Reply:** We understand your concern about the robustness of the regionalisation processes and their effect on the simulated hydrographs.

The analysis that you propose of comparing the regionalised parameters would be very interesting (and to our knowledge it would be novel) when using a regression-based method, where the parameters are estimated as a regression function of catchment characteristics and such function might allow to discuss the influence of regionalisation on the corresponding hydrographs: in such case we might expect to identify patterns of how the regionalised parameters vary across the study area, having a direct relationship with the attributes, which may contribute to systematic errors in the simulations.

But such comparison is not possible in our work because the regionalisation methods that we tested all belong to the "distance-based" class of approaches, which identify the model simulation at the target section based on the repetition of the simulation with one or multiple sets of parameters found for donor catchments (spatially closer in NN or most similar in MS) or based on the spatial correlation of the parameters (KR) and there is not a univocal influence of the regionalisation (or of the exclusion of nested donors or of the reduction of the density) on the parameters and therefore on the hydrographs.

We fully agree that reporting only goodness-of-fit indexes do not fully diagnose the capability of the model to accurately reproduce observations, but we did not show any hydrograph in the manuscript because we think that focusing on a single catchment could be misleading: the effect of the regionalisation, and the effect of the exclusion of nested catchments or of the density reduction on the approaches (that is our main focus) is strictly related to the single watershed and its donors, it varies based on the approaches and the specific topological relationships and characteristics of the catchment location.

As an example, we report below the analysis on the effect of the exclusion of nested donor basins when regionalising model parameters for one of the catchments in the Austrian dataset, and in particular we chose one target where excluding a nested donor resulted in a very strong deterioration for one of the methods, so that it is well visible on the hydrographs.

For the sake of brevity, the results regarding the only TUW model and the NN-1, MS-1 and KR regionalisation methods are reported. Moreover, just Criterion 2 is used for nested exclusion.

Figure R1.1 shows the position of the example target catchment $T$ (yellow) and the selected donor basins according to the NN-1 and MS-1 method: basin $A$ is the basin selected as donor for both methods NN-1 and MS-1 and it is nested to $T$. If we imagine to remove the availability of nested catchments, $A$ would be no longer available and the methods would select respectively basin $B$ (closest for NN-1) and $C$ (most similar for MS-1). Of course, this representation is not feasible for the kriging method KR since it regionalises each model parameter independently, based on the parameters of all donors, and it is not based on the selection of one or more specific donors.

[Figure]

Figure R1.1. Location of the donor catchments for an example target basin *T*: nested basin *A* is the selected donor catchment for both methods NN-1 and MS-1, while if nested are not considered (following Criterion 2) basin *B* and *C* are selected as donor respectively from NN-1 and MS-1 approaches.

Figure R.1.2 reports observed and simulated hydrographs in year 1997 for the three approaches when including or excluding (Criterion 2) the nested donors and Table R.1.1 reports the parameter sets "at site" and regionalised. It can be noticed that, for this specific catchment, the method that deteriorates more when excluding the nested donor is MS-1 (panel c) whose simulated hydrograph (red line) differs substantially from observation (grey points) and from the regionalised simulation obtained when using the nested donor (blue line) as well, while for NN-1 and KR simulations are just slightly affected by the exclusion of A from the potential donors. The reason behind the strong degradation of MS-1 performances can be found looking at the change in the parameter set when excluding the nested donor *A*: the at-site parameter sets of the target and of its nested donor *A* are quite similar, while the parameters of donor *B* differ substantially; the smaller value of FC and $k_1$, along with a lower threshold $L_{UZ}$, lead to small storage capacity and thus to much more higher flow peak (red line in Figure R.1.2, panel c) in disagreement with observation. This does not happen, in this example, for the remaining two approaches where the regionalised parameter sets are still quite similar to the "at site" calibrated one, even when *A* is excluded.

Table R1.1. Parameter sets for example target catchment *T*.

| | At Site | NN-1 Regionalized (no exclusion) | NN-1 Regionalized (Criterion 2) | MS-1 Regionalized (no exclusion) | MS-1 Regionalized (Criterion 2) | KR Regionalized (no exclusion) | KR Regionalized (Criterion 2) |
|---|---|---|---|---|---|---|---|
| SCF | 0.9 | 0.9 | 0.9 | 0.9 | 1.2 | 0.9 | 0.9 |
| DDF | 1.9 | 2.4 | 3.1 | 2.4 | 3.4 | 2.5 | 2.5 |
| LP | 0.2 | 0.0 | 1.0 | 0.0 | 1.0 | 0.4 | 0.7 |
| FC | 285.1 | 114.4 | 150.7 | 114.4 | 33.5 | 177.7 | 137.1 |
| $\beta$ | 1.0 | 20.0 | 20.0 | 20.0 | 1.3 | 15.7 | 13.2 |
| $k_0$ | 0.3 | 0.3 | 0.4 | 0.3 | 1.9 | 0.4 | 0.4 |
| $k_1$ | 16.1 | 6.3 | 8.0 | 6.3 | 2.3 | 9.0 | 12.5 |
| $k_2$ | 86.8 | 51.0 | 73.6 | 51.0 | 30.0 | 88.8 | 84.9 |
| $L_{UZ}$ | 42.8 | 65.0 | 87.1 | 65.0 | 18.8 | 71.4 | 76.3 |
| $C_{PERC}$ | 1.2 | 4.7 | 5.8 | 4.7 | 1.9 | 4.6 | 4.7 |
| $C_{ROUTE}$ | 49.7 | 19.3 | 43.5 | 19.3 | 16.0 | 32.6 | 43.9 |

[Figure]

Figure R1.2. Observed and simulated hydrographs in year 1997 for the example target catchment $T$: panel a) average catchment precipitation, panel b c and d) observed streamflow (grey points) and simulated streamflow (respectively for approaches NN-1, MS-1 and KR) when nested catchments are available (blue line) or not (red line).

The selected target catchment represents an unlucky case in which one of the method (MS-1) is deeply affected by the exclusion of the available nested donor, since the "most similar" among the not-nested donors, $C$, is evidently not so hydrologically similar to the target and its parameters do not allow a good simulation of the rainfall-runoff transformation in $T$. Anyway, we decided to display it to highlight the differences between the simulated hydrographs, otherwise not visible in less 'unlucky' cases.

A different catchment would include different multiple donors and different changes in the parameters and, consequently, in the hydrographs: the effect of the regionalisation and of the presence of nested donors is strictly related to the characteristic and the location of the single target catchment analysed.

(And it should also be noted that this example shows the behaviour for the single donor methods, but if we analysed the output averaging methods, we would need to take into account other 3 (for NN-OA) plus 3 (for MS-OA) sets of parameters (and other 3+3 when excluding the nested donors), thus making this kind of representation even longer).

For these reasons, we would prefer not to include specific examples in the manuscript.

**Response to Reviewer 2**

**General comments:**
The authors present a very good work on assessing the reliance of model regionalization approaches on the gauged data content. The methodology is clear and easy to follow. Their results indicate that transferring the entire parameter set, thus keeping the correlation among model parameters, outperforms the parameter-averaging kriging method. The output-averaging methods using more than one donor basins is more robust than the one-donor method. The most similar method based on geomorphological and climatic descriptors tends to show higher transferability in sparsely gauged areas than the nearest neighbor method. The findings provide important reference for the community when conducting hydrological modeling in sparsely or ungauged basins.

**A)** My major concern is that the structure of the manuscript is a bit scattered. The abstract and results sections are lengthy, from my taste. The authors could consider potential rephrase or reorganization. In addition, some Paragraphs in the results section should move to the methodology section.
**Reply:** As suggested, we have substantially cut and revised the abstract and moved some paragraphs (moved to lines 349-358, 363-379) from the results to the methodology section (see also reply to minor comments).

**B)** My second concern is that the applied two hydrological models have different calibrated/regionalized parameter spaces. In particular, the TUW model implies 11 parameters for calibration, while the GR6J model has less 8 parameters for calibration/regionalization. Some discussions on the effects of the size of parameter space on the results are needed.
**Reply:** The issue of the complexity/parsimony of the models is a very important topic: in our case, we assume that the complexity of the two models is similar, since the number of parameters is not so different (8 vs 11 parameters).
Previous studies in the literature applied parameter regionalisation to different conceptual rainfall-runoff models with different number of parameters: Chiew et al. (2010) and Petheram et al. (2012) implemented various models with parameter number varying between 6 and 14 without reporting a clear dependence of model performances on the dimension of parameter space; Viney et al. (2009) used models with 6 to 13 parameters and reported an increasing calibration performance for larger parameter space, but results did not show such a clear behaviour in regionalisation, where results were more dependent on the approach.
For what concern our study, in both results (old lines 382-383, 443-444, 538-540) and discussion (old lines 649-651, 665-666, 697-699) sessions we argued that GR6J model performances "at site" are slightly better than TUW despite the lower number of model parameters, while in general, for regionalisation, the TUW parameters seems to be easier to transfer from gauged to ungauged catchments (and this is even more accentuated when excluding nested donor catchments).
We think that this effect is probably not so much related to the number of parameters, but rather due the higher conceptualisation level of the GR6J model, that may lead to have some parameters that are very case-dependent, and are therefore related to physical basin characteristic (and thus to catchment similarity): for example, the groundwater module of GR6J includes an exchange function (not complying with the volume balance) based on two parameters X2 and X5 which are very sensitive to the specific local conditions and therefore likely difficult to transfer.
In conclusions of the revised version (lines 664-668) we have added, as suggested, that the results may depend on the different model structure and on the different transferability of model parameters (depending also on their meaning and their relation with the available catchment attributes). This aspect would deserve more attention and investigation but it would need a separate ad-hoc analysis, since the comparison of the structures and of the physical meaning of the parameters of the two models is not the specific objective of our work.

**C)** Third, the studied 209 catchments vary across a large range of area from 13 to 6000 km². But, the models didn't use distributed parameter values. That is, the spatial variability of parameters in large basins wasn't considered. The author should also add discussions on the performance of the regionalization approaches in catchments with distinct basin areas (small, moderate and large basin).

**Reply:** This is indeed a very interesting question, that we had not explicitly analysed yet. In general, evidence from the literature tends to show that regionalisation performance as well as "at site" performances (see, e.g. Merz et al., 2009; 2011; and Nester et al., 2011) improve for increasing catchment size. For instance, in the review of Parajka et al. (2013) the authors gather and compare the outcomes of different studies predicting runoff hydrographs through rainfall runoff model regionalisation, finding a clear pattern of an increase of model accuracy with catchment scale. As they say, this is generally due to two reasons. The first is a trend for an increasing number of raingauges within a catchment as the catchment size increases, which increase accuracy of rainfall input to models. The second may be related to the aggregation effect of runoff: as the catchment size increases, some of the hydrological variability is averaged out due to an interplay of space–time scale processes, which will improve hydrological simulation.

We have checked, as suggested by the referee, if the performance of the regionalisation methods is influenced by basin size: as suggested, we have divided the study sample in three equally numerous groups:

- Small (area <= 99 km², 70 catchments)
- Medium (99 km² < area <=285 km², 69 catchments)
- Large (area > 285 km², 70 catchments)

and we have checked for any change in regionalisation performance. Similarly to Figure 7 in the manuscript, Figure R.2.1 below shows KGE regionalisation performances for the three groups and the two models.

It may be seen that there are no strong differences between the three groups but, as expected, worse regionalisation accuracies occur in general for smaller catchments.

More interestingly, the ranking of the performances obtained by the different methods does not significantly change for different catchment size.

In the revised manuscript (lines 452-456) we have addressed the issue and specified that despite the different extension of the catchments in the dataset, even if worse performances tend to occur in general for smaller catchments, consistently with previous evidence from the literature (see, e.g. Parajka et al 2013), the efficiencies of the regionalisation methods (and their ranking) do not show a clear relation with the size of the watershed. And we have also specified (line 110) that in our study region catchment size ranges mainly between 13 and 1000 km² (90% of the basins) and just 3 watersheds extend over more than 3000 km² (we thank the referee for having highlighted the usefulness of such clarifications). We do not think that adding a new, large figure in the revised manuscript is needed (as highlighted by the referee we have already too many figures), but of this may be done if needed.

[Figure]

small basins medium basins large basins

Figure R2.1.

**D)** Finally, the authors should also specify how they sampled catchment subsets from the total 209 catchments when investigating the effects of station density. Did they sample manually or using automatic scripts? The thing here is how to guarantee that the sampled catchments are evenly distributed across the country. The sampled catchments concentrating in a small region could result in a same station density with the evenly distributed catchments.

**Reply:** Thank you for expressing your concern about this issue: we agree that it deserves additional clarification and further analysis.

We performed a simple automatic non-supervised sampling (now specified at line 353).

The chance that a cluster of catchments results to be sampled in a small region can not be excluded, even if it is unlikely. For this reason, the samplings are repeated 100 times for each value of station density, thus considering several groups of basins with the same density in the study region.

For addressing the point suggested by the referee, we have verified the distribution of our samples across the country, and we have added such new analysis to the revised version at lines 561-581.

In order to verify if the catchment samples are sufficiently evenly distributed across the country, let's consider a group of catchments and let's measure the distance of each catchment from its closest potential donor as shown the following figure:

**Distance from closest donor**

Figure R2.2. Example of distance from closest donor.

The average of the distances ($d_1$, $d_2$, $d_3$, $d_4$, $d_5$) of each catchment from the closest catchment (i.e. a potential donor) in a sample can be considered as a measure of the sample spatial distribution: the higher the distance the less dense the sample. As above said, for each density, 100 different samples are generated, so that for each density, we have 100 different values for such averages.

The figure below shows the average "distance within sample" of the closest available donor catchment across the 100 generated sub-sets for the different values of station density (each boxplot refers to the 100 values of average distance calculated for each sub-set). The average distance from the closest donor in the original, full density dataset (grey point in the figure) is around 8.5 km. As expected, the median target/donor distance (middle black solid line in each box) increases with decreasing density: it is true that also the variability of the distance, as shown by box size and whiskers, gradually increases with the reduction of station density, but such increase is overall modest: even for the lowest density, it is limited to +/- 18% of the median for the 80% of the samples. The fact that, on average, the distance between a target catchment and the closest gauged catchment consistently increases for decreasing density proves that the samples with lower density do not tend to cluster/concentrate the catchments in a small region, but there is an even distribution over the country.

[Figure]

Figure R.2.3. Boxplots of the average distance within sample from the closest available potential donor catchment across the 100 generated sub-sets, for different values of station density (gauges/1000km$^2$). Whiskers extend to 10th and 90th percentiles. The grey point indicates the average distance from the closest donor in the original dataset."

We are extremely grateful to the Referee for having suggested this interesting analysis, that we have added in the manuscript (section 4.4), since it is indeed a very important point.

**Specific Comments:**
**1.** Please, keep "HBV" and "TUW" consistent. Otherwise, the readers would find three hydrological models in this study. Actually, there are only two models.
**Reply:** Thank you for the correction. We have changed "HBV" to "TUW" when referring to the model used in this study.

**2.** Lines 8-10, consider to move to introduction section.
**Reply:** We think that it may be useful for introducing the topic to readers that may not be acquainted with the regionalisation of rainfall-runoff models, and we would prefer keeping this sentence in the abstract.

**3.** Lines 21-22, may consider to remove.
**Reply:** We agree, it has been removed in the revised abstract.

**4.** Lines 33-34, you already specified "how" above. Consider to remove.
**Reply:** We agree and have modified the abstract avoiding the repletion.

**5.** Line 67, "the best the best".
**Reply:** We corrected this typo.

**6.** Line 96, what do you mean by "continuous simulating daily models"?
**Reply:** We mean they are not event-based models. We have changed it to "continuous-simulation daily rainfall-runoff models" as used more frequently in hydrology (e.g., by Crooks and Naden 2007).

**7.** Line 116, "the more".
**Reply:** We have changed "to the more than" to "to more than".

**8.** Line 259, how to calculate "stream network density", please specify this

**Reply:** This attribute was computed in previous studies over the same data set (Section 2, l. 133-135), but we tried to explain it better in the revised manuscript (lines 130-131).

**9.** Lines 305-322, consider to move that to introduction. Here, you could add description on your method to choose the number of donor catchments.
**Reply:** We would prefer to keep this section here since it is specific for the choice of the number of donors, which is not the main issue of the manuscript. We are afraid it could divert the readers' attention from the main research focus if moved to introduction.

**10.** Line 289, by running.
**Reply:** We corrected it, thank you.

**11.** Lines 366-373, move to methodology section.
**Reply:** We agree: we moved it to a new methodology sub-section (lines 363-379).

**12.** Line 392, "as anticipated" → "as introduced".
**Reply:** We corrected it, thank you.

**13.** Line 402, "so relevant" → "so obvious".
**Reply:** We have rephrased the sentence.

**14.** Lines 425-426, 460-471, move to methodology.
**Reply:** The first lines have been moved to the new methodology section (along with old lines 366-373). We have also moved Figure 7 to section 3.3, but for what concerns lines 460-471, we consider the resulting sets of the nested catchments and the definition of the percentage threshold as part of the results, since they depend on the case study. Thus, we would prefer to keep them in section 4.3.1.

**15.** Line 507, KGE and NSE.
**Reply:** We have corrected it, thank you.

**16.** Lines 528-533, consider to rephrase that. Hard to follow.
**Reply:** We have simplified and divided the sentences, thank you.

**17.** Lines 538-540, could you add some discussions on that?
**Reply:** We believe it is related to the different structure and number/sensitivity of model parameters. As said above, it is of course a matter of strong interest and it would deserve more attention but we prefer to not include further discussion here since it is not the focus of the study. However, we will address again the issue in this section as well.

**18.** Lines 570-577, consider to move to methodology.
**Reply:** As prompted also by comment D, we agree that we need to better clarify the sampling procedure already in the methodology section and we have moved the description in Section 3.4.

**19.** Lines 685-687, please rephrase that, cannot follow.
**Reply:** The sentence has been removed to simplify the paragraph.

**20.** Please, consider to reduce the number of figures. Try to aggregate Figures 1 and 2. One of the options is try to present the performance of the two hydrological models in one figure instead of showing separately, such as figures 10-11, 12-13, 14-15. Or you may consider to provide the figures in a supplementary file.
**Reply:** As suggested, we have put together figures 1 and 2, as well as figures 10-11, 12-13 and 14-15. We would prefer to keep them in the main text, since they are needed to interpret the discussion and conclusions.

**References used in this response to the reviewer**

Chiew, F. H. S.: Lumped Conceptual Rainfall-Runoff Models and Simple Water Balance Methods: Overview and Applications in Ungauged and Data Limited Regions, Geogr. Compass, 4/3, 206–225, doi:10.1111/j.1749-8198.2009.00318.x, 2010.

Crooks, S. M. and Naden, P. S.: CLASSIC: a semi-distributed rainfall-runoff modelling system, Hydrol. Earth Syst. Sci., 11, 516–531, https://doi.org/10.5194/hess-11-516-2007, 2007.

Merz, R., Parajka, J., and Blöschl, G.: Scale effects in conceptual hydrological modeling, Water Resour. Res., 45, W09405, doi:10.1029/2009WR007872, 2009.

Merz, R., Parajka, J., and Blöschl, G.: Time stability of catchment model parameters: Implications for climate impact analyses, Water Resour. Res., 47, W02531, doi:10.1029/2010WR009505, 2011.

Nester, T., Kirnbauer, R., Gutknecht, D., and Blöschl, G.: Climate and catchment controls on the performance of regional flood simulations, J. Hydrol., 402, 340–356, 2011.

Parajka, J., Viglione, A., Rogger, M., Salinas, J. L., Sivapalan, M., and Blöschl, G.: Comparative assessment of predictions in ungauged basins – Part 1: Runoff-hydrograph studies, Hydrol. Earth Syst. Sci., 17, 1783–1795, https://doi.org/10.5194/hess-17-1783-2013, 2013.

Petheram, C., Rustomji, P., Chiew, F. H. S., and Vleeshouwer, J.: Rainfall-runoff modelling in northern Australia: a guide to modelling strategies in the tropics, J. Hydrol., 462–463, 28–41, https://doi.org/10.1016/j.jhydrol.2011.12.046, 2012.

Viney, N. R., Perraud, J. Vaze, J., Chiew, F. H. S., Post, D. A., and Yang, A.: The usefulness of bias constraints in model calibration for regionalisation to ungauged catchments, in: Proceed, 18th World IMACS/MODSIM Congress, Cairns, Australia 13–17 July, 2009.

[revised manuscript text omitted]

**Commentato [MN4]:** Modified based on comment 3 of Reviewer 1

**Commentato [MN5]:** Added based on comment 1 of Reviewer 1

performances for a given approach, but such accuracy may not represent what would be obtained in different conditions. Therefore, regionalisation performances obtained for datasets with high degree of "nestedness" may be not transferrable to study regions poor of nested basins.

So far, very few studies have been presented in the literature regarding the impact of the presence of nested catchments on the performances of parameter regionalisation techniques. Merz and Blöschl (2004), Parajka et al. (2005) and Oudin et al. (2008) tested the effect of the removal of nested catchments from the available donor catchments, but only for one or two regionalisation techniques, without analysing in detail the differences between different types of approaches. Additionally, the contribute of the immediate downstream and/or upstream gauged stations has never been compared to that of the remaining nested catchments, that may share significant portions of drainage area with the ungauged one.

Also the influence of the density of the gauging stations on the parameterisation of rainfall-runoff models has been little explored, with two notable exceptions: Oudin et al. (2008) applied the spatial proximity and physical similarity output-averaging techniques for decreasing values of station density in France and Lebecherel et al. (2016) tested the robustness of the spatial proximity output-averaging approach to an increasing sparse hydrometric network on the same study region. In Austria, the effect of station density has been investigated by Parajka et al. (2015), but in reference to the interpolation of streamflow time-series and not to the parameterisation of rainfall-runoff models.

The purpose of the present paper is to analyse the role of the informative content of the available regional data set, that is which and how many gauged catchments are available to be used as donors for the regionalisation in a target, ungauged section. This will be done comparing first the impact of the presence of nested donors and then the effect of the reduction of station density on the performances of different parameter regionalisation techniques for a dataset of 209 catchments across Austria.

The tested regionalisation approaches include a set of consolidated techniques, applied to two different continuous-simulation daily rainfall-runoff models, for generalisation purposes: the first is the TUW model (semi-distributed version of HBV, used by Parajka et al. 2005), and the second model, never used so far for regionalisation in the Austrian region, is the GR6J model implemented with the Cemaneige snow routine (Coron et al., 2017b).

We believe that the present analysis may provide further insights for assessing the performances and selecting the parameter regionalisation approaches most suitable to a specific study region, keeping into account the impact of data availability, and in particular of gauging density and of the presence of nested catchments.

The paper is organised as follows: Section 2 introduces the case study and data. Section 3 first describes the rainfall-runoff models and the tested regionalisation schemes, then the methodology for assessing the impact of nested catchments and of station density is presented, while the results are presented in Section 4. Finally, Section 5 reports the discussion and the conclusions.

**2 Study region and data**

The case study is composed by 209 catchments (see Figure 1, panel a) covering a large portion of Austria. Their size varies considerably, mainly under 1000 km$^2$ (90% of the basins) and just 3 watersheds extend over more than 3000 km$^2$. The topography of the country varies significantly from the flat and hilly area in the north-east to the Alps in the centre and in the south-west, particularly steep in the extreme west. The annual precipitation ranges from about 600 mm in the

**Commentato [MN6]:** This paragraph was extended based on comment 1 of Reviewer 1

**Commentato [MN7]:** Modified based on specific comment 6 of Reviewer 2

**Commentato [MN8]:** Modified based on comment C of Reviewer 2

[revised manuscript text omitted]

---

## Author Response (AR2)

**Response to Anonymous Referee #2**

I appreciate the authors' work on addressing my concerns. The paper is now much improved. However, the English writing is still poor. I would suggest the senior co-authors to play some efforts to refine/smooth the writing in a revised version. The authors used the symbol ":" very often to construct rather long sentences based on unclear logics, which however is very unfriendly to the readers. Beyond that, grammar errors as well as sloppy long sentences make the texts hard to follow. Some examples are listed below, but not limited to this. Further smoothed writing by an English native speaker would be suggested.
**Reply:** We thank the referee for the precious and accurate corrections and suggestions. We have included most of them in the text and we tried to further improve the English making it flowing better and paying particular attention to long sentences, which were substantially cut and revised. The new changes in the manuscript are in blue text (please see the marked-up version of the manuscript attached below).

Line 10, activities;
**Reply:** The error was corrected.

line 13, why is the ":" used here?;
**Reply:** We used a colon because our work stems from the previous consideration (the fact that it is important to better understand the role of the informative content). But in order to make the paragraph shorter, we replaced ":" with ".".

line 20, "consists in"- is on the effects of; line 21, "taking into account"-based on;
**Reply:** We rephrased the entire sentence.

line 22, "secondly"- second/moreover;
**Reply:** We changed the text as suggested, thank you.

line 25, "the 'output-averaging'…data-rich regions". Please rephrase;
**Reply:** The sentence, that was indeed too convoluted (we thank you for pointing it out) was rephrased and divided into two periods.

line 31, ways; line 61, "result"- be obtained; line 65, "for"- of; 126, delete "reported and";
**Reply:** We corrected/changed the text as suggested, thank you.

line 98, delete "for generalization purposes"; line
**Reply:** We would keep it, it explains why we use two models instead than one.

line 130, please specify how did you calculate the stream network density, is it by the ratio between the channel area and the catchment area?
**Reply:** Yes, we have now specified it.

lines 135-137, please rephrase;
**Reply:** The sentence was rephrased.

line 141, patterns;
**Reply:** We corrected it, thank you.

line 143, Table 1. Catchment characteristics used in the parameter regionalisation approaches;
**Reply:** We prefer to keep the caption as it is because just five of these descriptors are finally used in the MS-1 and MS-OA approaches.

line 151, "consists in"- consists of; line 155, based on the sub-catchment areas; line 158, "routine"- module, same for the follows; line 170, by dividing; line 175, Table 2 describes the physical basis and the value ranges of the calibrated parameters;
**Reply:** We corrected/changed the text as suggested, but putting at previous line 175: "presents the parameters to be calibrated and the corresponding ranges".

line 202, "elevation layer"- elevation band;
**Reply:** This is the term used in the Valery et al 2014, so we'd prefer to keep it as named by the authors.

line 207, provide inputs for the GR6J;
**Reply:** We corrected/changed the text as suggested, thank you.

line 208, delete "accounting"; line 209, routing procedure of the module includes two flow components; line 212, fed by;
**Reply:** We corrected/changed the text as suggested, thank you.

lines 214-215, please rephrase;
**Reply:** The sentence was changed, removing some useless detail, thank you for pointing this out.

line 216, reports briefly, for the sake of brevity; line 218, and their value ranges;
**Reply:** We corrected/changed the text as suggested, thank you.

line 224, using the algorithm of Dynamically Dimensioned Search;
**Reply:** We amended the sentence.

line 230, is r the Pearson correlation coefficient or the linear correlation coefficient? Please have a double check;
**Reply:** Yes, it is the Pearson product-moment correlation coefficient line 239, demonstrated how? line 280, demonstrating how?
**Reply:** We corrected them with "demonstrated that", at both lines.

Line 249, "complete"- entire;
**Reply:** We corrected it, thank you.

Lines 282-285, please rephrase; lines 441-442, please rephrase that;
**Reply:** The sentences were rephrased.

line 287, 1:n; line 322, one of the main purposes; line 364, as mentioned above; line 382, obtained by calibrating; line 406, Results show that in all the four cases; line 475, delete "the";
**Reply:** We corrected/changed the text as suggested, thank you.

line 420, is reduced;
**Reply:** We used "dampened" instead of reduced, for replacing smoothed line 455, performance is lower in smaller catchments is of course a clear relation;
**Reply:** This is not exactly what we mean: we do not see a clear relation, but we just notice that if it happens that performance is not very good, it is in general for small catchments (but most of small catchments still behave well). For this reason we replaced the sentence with "even if for some of the smaller catchments the performances were suboptimal"

lines 476-479, please rephrase; lines 529-531, please rephrase;
**Reply:** We rephrased them.

line 483, 43% of that have;
**Reply:** We used just "43% have" since we have just above cited the set to which the percentage refers.

line 525, delete "as";
**Reply:** We rephrased the sentence, to clarify it.

line 531, bottom panel;
**Reply:** We kept the plural since we refer to all the 4 bottom panels of figure 10

line 540, as already mentioned; line 573, increases with decreasing; line 574, but are evenly distributed over the country; line 584, consists of;
**Reply:** We corrected/changed the text as suggested, thank you.

line 570, it is also true with respective to the variability of:
**Reply:** We rephrased the sentence line 595, delete "large";
**Reply:** We'd prefer to keep it.

line 608, this also holds true; line 621, reduces from; line 629, impacts; line 631, effects of data-richness and informative content on the accuracy of various methods; line 634, "extendable"- transferable; line 664, structures; line 675, "closeness"- proximity; line 708, is more noticeable; line 712, as in Austria;
**Reply:** We corrected/changed the text as suggested, thank you.

lines 637-640, please rephrase;
**Reply:** The sentence was split and rephrased.

lines 646-653, please rephrase; lines 656-662, please rephrase;
**Reply:** We have rephrased these paragraphs.

lines 695-698, please rephrase;
**Reply:** The sentence was modified.

lines 715-720, please rephrase;
**Reply:** The sentences were split and modified.

[revised manuscript text omitted]